# Stochastic Continuous Greedy ++:
# When Upper and Lower Bounds Match*

**Hamed Hassani**
ESE Department
University of Pennsylvania
Philadelphia, PA
hassani@seas.upenn.edu

**Amin Karbasi**
ECE Department
Yale University
New Haven, CT
amin.karbasi@yale.edu

**Aryan Mokhtari**
ECE Department
The University of Texas at Austin
Austin, TX
mokhtari@austin.utexas.edu

**Zebang Shen**
ESE Department
University of Pennsylvania
Philadelphia, PA
zebang@seas.upenn.edu

## Abstract

In this paper, we develop `Stochastic Continuous Greedy++` (SCG++), the first efficient variant of a conditional gradient method for maximizing a continuous submodular function subject to a convex constraint. Concretely, for a monotone and continuous DR-submodular function, SCG++ achieves a tight $[(1-1/e)\text{OPT} - \epsilon]$ solution while using $O(1/\epsilon^2)$ stochastic gradients and $O(1/\epsilon)$ calls to the linear optimization oracle. The best previously known algorithms either achieve a suboptimal $[(1/2)\text{OPT} - \epsilon]$ solution with $O(1/\epsilon^2)$ stochastic gradients or the tight $[(1-1/e)\text{OPT} - \epsilon]$ solution with suboptimal $O(1/\epsilon^3)$ stochastic gradients. We further provide an information-theoretic lower bound to showcase the necessity of $\mathcal{O}(1/\epsilon^2)$ stochastic oracle queries in order to achieve $[(1-1/e)\text{OPT} - \epsilon]$ for monotone and DR-submodular functions. This result shows that our proposed SCG++ enjoys optimality in terms of both approximation guarantee, i.e., $(1-1/e)$ approximation factor, and stochastic gradient evaluations, i.e., $O(1/\epsilon^2)$ calls to the stochastic oracle. By using stochastic continuous optimization as an interface, we also show that it is possible to obtain the $[(1-1/e)\text{OPT} - \epsilon]$ tight approximation guarantee for maximizing a monotone but stochastic submodular set function subject to a general matroid constraint after at most $\mathcal{O}(n^2/\epsilon^2)$ calls to the stochastic function value, where $n$ is the number of elements in the ground set.

## 1 Introduction

In this paper, we consider the following *non-oblivious* stochastic submodular maximization problem:

$$\max_{\mathbf{x} \in \mathcal{C}} F(\mathbf{x}) := \max_{\mathbf{x} \in \mathcal{C}} \mathbb{E}_{\mathbf{z} \sim p(\mathbf{z};\mathbf{x})}[\tilde{F}(\mathbf{x}; \mathbf{z})], \tag{1}$$

where $\mathbf{x} \in \mathbb{R}_+^d$ is the decision variable, $\mathcal{C} \subseteq \mathbb{R}^d$ is a convex feasible set, $\mathbf{z} \in \mathcal{Z}$ is a random variable with distribution $p(\mathbf{z}; \mathbf{x})$, and the submodular objective function $F : \mathbb{R}^d \to \mathbb{R}$ is defined as the expectation of a set of stochastic functions $\tilde{F} : \mathbb{R}^d \times \mathcal{Z} \to \mathbb{R}$. In this paper, we focus on a general case of stochastic submodular maximization in which the probability distribution of the random variable $\mathbf{z}$ depends on the variable $\mathbf{x}$ and may change during the optimization procedure. One should

note that the usual stochastic optimization where the distribution $p$ is independent of $\mathbf{x}$ is a special case of Problem (1). A canonical example of the general stochastic submodular maximization problem in (1) is the multi-linear extension of a discrete submodular function where the stochasticity crucially depends on the decision variable $\mathbf{x}$ at which we evaluate. Specifically, consider a discrete submodular set function $f : 2^V \rightarrow \mathbb{R}_+$ defined over the ground set $V$. The aim is to solve the following problem $\max_{S \in \mathcal{I}} f(S)$ where $\mathcal{I}$ is a matroid constraint. For this problem, the classic greedy algorithm leads to a $1/2$ approximation guarantee, but one can achieve the optimal approximation guarantee of $1 - 1/e$ by maximizing its multilinear extension $F : [0, 1]^V \rightarrow \mathbb{R}_+$ which is defined as

$$F(\mathbf{x}) \ := \ \mathbb{E}_{\mathbf{z} \sim \mathbf{x}}[f(\mathbf{z}(\mathbf{x}))] \ := \ \sum_{S \subseteq V} f(S) \prod_{i \in S} x_i \prod_{j \notin S} (1 - x_j), \tag{2}$$

where for the random set $\mathbf{z}(\mathbf{x})$ each element $e$ is sampled with probability $x_e$. This problem is a special case of (1) if we define $\tilde{F}(\mathbf{x}, \mathbf{z})$ as $f(\mathbf{z}(\mathbf{x}))$ and $p(\mathbf{x}, \mathbf{z})$ as the distribution of the random set $\mathbf{z}(\mathbf{x})$, i.e., each coordinate $z_e$ is generated according to a Bernoulli distribution with parameter $x_e$.

When $F$ is monotone and continuous DR-submodular, Hassani et al. [17] showed that the Stochastic Gradient Ascent (SGA) method finds a solution to Problem (1) with a function value no less than $[(1/2)\text{OPT} - \epsilon]$ after computing $\mathcal{O}(1/\epsilon^2)$ stochastic gradients. Here, and throughout the paper, OPT denotes the optimal value of Problem (1). Hassani et al. [17] also provided examples for which SGA cannot achieve better than $1/2$ approximation ratio, in general. Later, Mokhtari et al. [22] proposed Stochastic Continuous Greedy (SCG), a conditional gradient method that achieves the tight $[(1 - 1/e)\text{OPT} - \epsilon]$ solution by $\mathcal{O}(1/\epsilon^3)$ calls to the linear optimization oracle while using $\mathcal{O}(1/\epsilon^3)$ stochastic gradients. While both SCG and SGA are first-order methods, meaning that they rely on stochastic gradients, SCG provably achieves a better result at the price of being slower. Therefore, a fundamental question is the following

> "*Can we achieve the best of both worlds? That is, can we find a* $[(1-1/e)OPT-\epsilon]$ *solution after at most* $\mathcal{O}(1/\epsilon^2)$ *calls to the stochastic oracle?*"

Another question that naturally arises is about a lower bound on the number of stochastic gradient evaluations for finding a $(1 - 1/e)$ approximate solution:

> "*What is the lower bound on the number of calls to the first-order stochastic oracle for achieving a* $[(1 - 1/e)OPT - \epsilon]$ *solution?*"

In this paper, we develop a tight lower bound on the number of calls to the stochastic oracle for achieving a $[(1-1/e)\text{OPT}-\epsilon]$ solution, and propose an algorithm that achieves the sample complexity of the lower bound. The detail of our contributions follows.

**Our contributions.** We develop `Stochastic Continuous Greedy++` (SCG++ ), the first method that achieves the tight $[(1 - 1/e)\text{OPT} - \epsilon]$ solution for Problem (1) with $\mathcal{O}(1/\epsilon)$ calls to the linear optimization program while using $\mathcal{O}(1/\epsilon^2)$ stochastic gradients in total. Our technique relies on a novel variance reduction method that estimates the difference of gradients in the non-oblivious stochastic setting without introducing extra bias. This is crucial in our analysis, as all the existing variance reduction methods fail to correct for this bias and can only operate in the oblivious/classic stochastic setting. We further show that our result is *optimal in all aspects*. In particular, we provide an information-theoretic lower bound to showcase the necessity of $\mathcal{O}(1/\epsilon^2)$ stochastic oracle queries in order to achieve $[(1 - 1/e)\text{OPT} - \epsilon]$. Note that under standard assumptions, one cannot achieve an approximation ratio better than $(1 - 1/e)$ for submodular functions [13]. By using stochastic continuous optimization as an interface, we also provide a $(1 - 1/e)\text{OPT} - \epsilon$ tight approximation guarantee for maximizing a monotone but stochastic *submodular set function* subject to a matroid constraint with at most $\mathcal{O}(n/\epsilon^2)$ calls to the stochastic oracle where $n$ is the size of the ground set.

## 2 Related Work

Submodular set functions capture the intuitive notion of diminishing returns and have become increasingly important in various machine learning applications. Examples include data summarization [20, 21], dictionary learning [9], and variational inference [11], to name a few. It is known that for a monotone submodular function and subject to a cardinality constraint, greedy algorithm achieves the tight $(1 - 1/e)$ approximation guarantee [25]. However, the vanilla greedy method does not provide

the tightest guarantees for many classes of feasibility constraints. To circumvent this issue, the continuous relaxation of submodular functions, through the multilinear extension, have been extensively studied [31, 7, 8, 14, 16, 30]. In particular, it is known that the Continuous Greedy algorithm achieves the tight $(1 - 1/e)$ approximation guarantee for monotone submodular functions under a general matroid constraint [7] with a prohibitive query complexity of $\mathcal{O}(n^8)$. The fastest existing solution for maximizing a submodular function subject to a matroid constraint interplays between discrete and continuous domains to achieve a running time of $\mathcal{O}(n/\epsilon^4)$ for finding a $(1 - 1/e)\text{OPT} - \epsilon$ approximate solution [4]. In contrast, we develop a pure continuous method that obtains the same guarantee with a running time of $\mathcal{O}(n^2/\epsilon^2)$.

Continuous DR-submodular functions, an important subclass of non-convex functions, generalize the notion of diminishing returns to the continuous domains [5]. Such functions naturally arise in machine learning applications such as Map inference for Determinantal Point Processes [19] and revenue maximization [26]. It has been recently shown that monotone continuous DR-submodular functions can be (approximately) maximized over convex bodies using first-order methods [5, 17, 22]. When exact gradient information is available, [5] showed that the continuous greedy algorithm achieves $[(1 - 1/e)\text{OPT} - \epsilon]$ with $O(1/\epsilon)$ gradient evaluations. However, the problem becomes considerably more challenging when we only have access to a *stochastic* first-order oracle. In particular, Hassani et al. [17] showed that the stochastic gradient ascent achieves $[1/2\text{OPT} - \epsilon]$ by using $O(1/\epsilon^2)$ stochastic gradients. In contrast, [22, 23] proposed a stochastic variant of continuous greedy that achieves $[(1 - 1/e)\text{OPT} - \epsilon]$ by using $O(1/\epsilon^3)$ stochastic gradients. This paper shows how to achieve $[(1 - 1/e)\text{OPT} - \epsilon]$ by $O(1/\epsilon^2)$ stochastic gradient evaluations.

## 3  Preliminaries

**Submodularity.**  A set function $f : 2^V \to \mathbb{R}_+$, defined on the ground set $V$, is submodular if $f(A) + f(B) \geq f(A \cap B) + f(A \cup B)$, for all subsets $A, B \subseteq V$. Even though submodularity is mostly considered on discrete domains, the notion can be naturally extended to arbitrary lattices [15]. To this aim, let us consider a subset of $\mathbb{R}_+^d$ of the form $\mathcal{X} = \prod_{i=1}^d \mathcal{X}_i$ where each $\mathcal{X}_i$ is a compact subset of $\mathbb{R}_+$. A function $F : \mathcal{X} \to \mathbb{R}_+$ is *continuous submodular* if $\forall (\mathbf{x}, \mathbf{y}) \in \mathcal{X} \times \mathcal{X}$

$$F(\mathbf{x}) + F(\mathbf{y}) \geq F(\mathbf{x} \vee \mathbf{y}) + F(x \wedge \mathbf{y}), \tag{3}$$

where $\mathbf{x} \vee \mathbf{y} \doteq \max(\mathbf{x}, \mathbf{y})$ (component-wise) and $\mathbf{x} \wedge \mathbf{y} \doteq \min(\mathbf{x}, \mathbf{y})$ (component-wise). A submodular function is monotone if for any $\mathbf{x}, \mathbf{y} \in \mathcal{X}$ such that $\mathbf{x} \leq \mathbf{y}$, we have $F(\mathbf{x}) \leq F(\mathbf{y})$ (here, by $\mathbf{x} \leq \mathbf{y}$ we mean that every element of $\mathbf{x}$ is less than that of $\mathbf{y}$). When twice differentiable, $F$ is submodular if and only if all cross-second-derivatives are non-positive [3], i.e., we have $\forall i \neq j, \forall \mathbf{x} \in \mathcal{X}, \ \partial^2 F(\mathbf{x})/\partial x_i \partial x_j \leq 0$. This expression shows continuous submodular functions are not convex nor concave in general, as concavity (convexity) implies that $\nabla^2 F \preceq 0$ (resp. $\nabla^2 F \succeq 0$). A proper subclass of submodular functions are called *DR-submodular* [29] if for all $\mathbf{x}, \mathbf{y} \in \mathcal{X}$ such that $\mathbf{x} \leq \mathbf{y}$ and any standard basis vector $\mathbf{e}_i \in \mathbb{R}^n$ and a non-negative number $z \in \mathbb{R}_+$ such that $z\mathbf{e}_i + \mathbf{x} \in \mathcal{X}$ and $z\mathbf{e}_i + \mathbf{y} \in \mathcal{X}$, then, $F(z\mathbf{e}_i + \mathbf{x}) - F(\mathbf{x}) \geq F(z\mathbf{e}_i + \mathbf{y}) - F(\mathbf{y})$. One can easily verify that for a differentiable DR-submodular function the gradient is an antitone mapping, i.e., for all $\mathbf{x}, \mathbf{y} \in \mathcal{X}$ such that $\mathbf{x} \leq \mathbf{y}$ we have $\nabla F(\mathbf{x}) \geq \nabla F(\mathbf{y})$ [5].

**Variance Reduction.**  Beyond the vanilla stochastic gradient, variance reduced methods [28, 18, 10, 27, 2] have succeeded in reducing stochastic first-order oracle complexity in *oblivious* stochastic optimization

$$\max_{\mathbf{x} \in \mathcal{C}} F(\mathbf{x}) := \max_{\mathbf{x} \in \mathcal{C}} \mathbb{E}_{\mathbf{z} \sim p(\mathbf{z})} \tilde{F}(\mathbf{x}; \mathbf{z}), \tag{4}$$

where each component function $\tilde{F}(\cdot; \mathbf{z})$ is *L-smooth*. In contrast to (1), the underlying distribution $p$ of (4) is invariant to the variable $\mathbf{x}$ and is hence called oblivious. We will now explain a recent variance reduction technique for solving (4) using stochastic gradient information. Consider the following *unbiased* estimate of the gradient at the current iterate $\mathbf{x}^t$:

$$\mathbf{g}^t := \mathbf{g}^{t-1} + \nabla \tilde{F}(\mathbf{x}^t; \mathcal{M}) - \nabla \tilde{F}(\mathbf{x}^{t-1}; \mathcal{M}), \tag{5}$$

where $\nabla \tilde{F}(\mathbf{y}; \mathcal{M}) := \frac{1}{|\mathcal{M}|} \sum_{\mathbf{z} \in \mathcal{M}} \nabla \tilde{F}(\mathbf{y}; \mathbf{z})$ for some $\mathbf{y} \in \mathbb{R}^d$, $\mathbf{g}^{t-1}$ is an unbiased gradient estimator at $\mathbf{x}^{t-1}$, and $\mathcal{M}$ is a mini-batch of random samples drawn from $p(\mathbf{z})$. [12] showed that, with the gradient estimator (5), $\mathcal{O}(1/\epsilon^3)$ stochastic gradient evaluations are sufficient to find an $\epsilon$-first-order stationary point of Problem (4), improving upon the $\mathcal{O}(1/\epsilon^4)$ complexity of SGD. A crucial

property leading to the success of the variance reduction method given in (5) is that $\nabla \tilde{F}(\mathbf{x}^t; \mathcal{M})$ and $\nabla \tilde{F}(\mathbf{x}^{t-1}; \mathcal{M})$ use *the same* minibatch sample $\mathcal{M}$ in order to exploit the $L$-smoothness of component functions $f(\cdot; \mathbf{z})$. Such construction is only possible in the oblivious setting where $p(\mathbf{z})$ is independent of the choice of $\mathbf{x}$, and would introduce bias in the more general non-oblivious case (1). To see this, let $\mathcal{M}$ be the minibatch of random variable $\mathbf{z}$ sampled according to distribution $p(\mathbf{z}; \mathbf{x}^t)$. We have $\mathbb{E}[\nabla \tilde{F}(\mathbf{x}^t; \mathcal{M})] = \nabla F(\mathbf{x}^t)$ but $\mathbb{E}[\nabla \tilde{F}(\mathbf{x}^{t-1}; \mathcal{M})] \neq \nabla F(\mathbf{x}^{t-1})$ since the distribution $p(\mathbf{z}; \mathbf{x}^{t-1})$ is not the same as $p(\mathbf{z}; \mathbf{x}^t)$. The same argument renders all the existing variance reduction techniques inapplicable for the non-oblivious setting of Problem (1).

## 4 Stochastic Continuous Greedy++

In this section, we present the `Stochastic Continuous Greedy++` (SCG++) algorithm which is the first method to obtain a $[(1 - 1/e)\text{OPT} - \epsilon]$ solution with $O(1/\epsilon^2)$ stochastic oracle complexity. The SCG++ algorithm essentially operates in a conditional gradient manner. To be more precise, at each iteration $t$, given a gradient estimator $\mathbf{g}^t$, SCG++ solves the subproblem

$$\mathbf{v}^t = \operatorname*{argmax}_{\mathbf{v} \in \mathcal{C}} \langle \mathbf{v}, \mathbf{g}^t \rangle \tag{6}$$

to obtain an element $\mathbf{v}^t$ in $\mathcal{C}$ as ascent direction, which is then added to the iterate $\mathbf{x}^{t+1}$ with a scaling factor $1/T$, i.e., the new iterate $\mathbf{x}^{t+1}$ is computed by following the update

$$\mathbf{x}^{t+1} = \mathbf{x}^t + \frac{1}{T}\mathbf{v}^t, \tag{7}$$

where $T$ is the total number of iterations of the algorithm. The iterates are assumed to be initialized at the origin which may not belong to the feasible set $\mathcal{C}$. Though each iterate $\mathbf{x}^t$ may not necessarily be in $\mathcal{C}$, the feasibility of the final iterate $\mathbf{x}^T$ is guaranteed by the convexity of $\mathcal{C}$. Note that the iterate sequence $\{\mathbf{x}^s\}_{s=0}^T$ can be regarded as a path from the origin (as we manually force $\mathbf{x}^0 = 0$) to some feasible point in $\mathcal{C}$. The key idea in SCG++ is to exploit the high correlation between the consecutive iterates originated from the $\mathcal{O}(1/T)$-sized increments to maintain a highly accurate estimate $\mathbf{g}^t$, which is the focus of the rest of this section. Note that by replacing the gradient approximation vector $\mathbf{g}^t$ in the update of SCG++ by the exact gradient of the objective function, we recover the update rule of the continuous greedy method [7, 5].

We now proceed to describe our approach for evaluating the gradient approximation $\mathbf{g}^t$ when we face a non-oblivious problem as in (1). Given a sequence of iterates $\{\mathbf{x}^s\}_{s=0}^t$, the gradient of the objective function $F$ at the iterate $\mathbf{x}^t$ can be written in a path-integral form as

$$\nabla F(\mathbf{x}^t) = \nabla F(\mathbf{x}^0) + \sum_{s=1}^t \left\{ \Delta^s \overset{\text{def}}{=} \nabla F(\mathbf{x}^s) - \nabla F(\mathbf{x}^{s-1}) \right\}. \tag{8}$$

By obtaining an unbiased estimate of $\Delta^t = \nabla F(\mathbf{x}^t) - \nabla F(\mathbf{x}^{t-1})$ and reusing the previous unbiased estimates for $s < t$, we obtain recursively an unbiased estimator of $\nabla F(\mathbf{x}^t)$ which has a reduced variance. Estimating $\nabla F(\mathbf{x}^s)$ and $\nabla F(\mathbf{x}^{s-1})$ separately as suggested in (5) would cause the bias issue in the the non-oblivious case (see discussion at the end of section 3). Therefore, we propose an approach for *directly estimating the difference* $\Delta^t = \nabla F(\mathbf{x}^t) - \nabla F(\mathbf{x}^{t-1})$ in an unbiased manner.

We construct an unbiased estimator $\mathbf{g}^t$ of the gradient vector $\nabla F(\mathbf{x}^t)$ by adding an unbiased estimate $\tilde{\Delta}^t$ of the gradient difference $\Delta^t = \nabla F(\mathbf{x}^t) - \nabla F(\mathbf{x}^{t-1})$ to $\mathbf{g}^{t-1}$, where $\mathbf{g}^{t-1}$ as an unbiased estimate of $\nabla F(\mathbf{x}^{t-1})$. Note that $\Delta^t = \nabla F(\mathbf{x}^t) - \nabla F(\mathbf{x}^{t-1})$ can be written as

$$\Delta^t = \int_0^1 \nabla^2 F(\mathbf{x}(a))(\mathbf{x}^t - \mathbf{x}^{t-1})\mathrm{d}a = \left[ \int_0^1 \nabla^2 F(\mathbf{x}(a))\mathrm{d}a \right] (\mathbf{x}^t - \mathbf{x}^{t-1}), \tag{9}$$

where $\mathbf{x}(a) \overset{\text{def}}{=} a \cdot \mathbf{x}^t + (1-a) \cdot \mathbf{x}^{t-1}$ for $a \in [0,1]$. Therefore, if we sample the parameter $a$ uniformly at random from the interval $[0,1]$, it can be easily verified that $\tilde{\Delta}^t := \nabla^2 F(\mathbf{x}(a))(\mathbf{x}^t - \mathbf{x}^{t-1})$ is an unbiased estimator of the gradient difference $\Delta^t$ since

$$\mathbb{E}_a[\nabla^2 F(\mathbf{x}(a))(\mathbf{x}^t - \mathbf{x}^{t-1})] = \nabla F(\mathbf{x}^t) - \nabla F(\mathbf{x}^{t-1}). \tag{10}$$

Therefore, all we need is an unbiased estimator of the Hessian-vector product $\nabla^2 F(\mathbf{y})(\mathbf{x}^t - \mathbf{x}^{t-1})$ for the non-oblivious objective $F$ at an arbitrary $\mathbf{y} \in \mathcal{C}$. In the following lemma, we present an unbiased estimator of $\nabla^2 F(\mathbf{y})$ for any $\mathbf{y} \in \mathcal{C}$ that can be evaluated efficiently.

---

**Algorithm 1** `Stochastic Continuous Greedy++` (SCG++)

---

**Input:** Minibatch size $|\mathcal{M}_0|$ and $|\mathcal{M}|$, and total number of rounds $T$
 1: Initialize $\mathbf{x}^0 = 0$;
 2: **for** $t = 1$ **to** $T$ **do**
 3:     **if** $t = 1$ **then**
 4:         Sample a minibatch $\mathcal{M}_0$ of $\mathbf{z}$ according to $p(\mathbf{z}; \mathbf{x}^0)$ and compute $\mathbf{g}^0 \overset{\text{def}}{=} \nabla \tilde{F}(\mathbf{x}^0; \mathcal{M}_0)$;
 5:     **else**
 6:         Sample a minibatch $\mathcal{M}$ of $\mathbf{z}$ according to $p(\mathbf{z}; \mathbf{x}(a))$ where $a$ is a chosen uniformly at
         random from $[0, 1]$ and $\mathbf{x}(a) := a \cdot \mathbf{x}^t + (1 - a) \cdot \mathbf{x}^{t-1}$;
 7:         Compute the Hessian approximation $\tilde{\nabla}_t^2$ corresponding to $\mathcal{M}$ according to (12);
 8:         Construct $\tilde{\Delta}^t$ based on (13) (Option I) or (18) (Option II);
 9:         Update the stochastic gradient approximation $\mathbf{g}^t := \mathbf{g}^{t-1} + \tilde{\Delta}^t$;
10:     **end if**
11:     Compute the ascent direction $\mathbf{v}^t := \text{argmax}_{\mathbf{v} \in \mathcal{C}} \{\mathbf{v}^\top \mathbf{g}^t\}$;
12:     Update the variable $\mathbf{x}^{t+1} := \mathbf{x}^t + 1/T \cdot \mathbf{v}^t$;
13: **end for**

---

**Lemma 1.** *For any* $\mathbf{y} \in \mathcal{C}$*, let* $\mathbf{z}$ *be the random variable with distribution* $p(\mathbf{z}; \mathbf{y})$ *and define*

$$
\begin{aligned}
\tilde{\nabla}^2 F(\mathbf{y}; \mathbf{z}) \overset{\text{def}}{=} &\tilde{F}(\mathbf{y}; \mathbf{z})[\nabla \log p(\mathbf{z}; \mathbf{y})][\nabla \log p(\mathbf{z}; \mathbf{y})]^\top + [\nabla \tilde{F}(\mathbf{x}; \mathbf{z})][\nabla \log p(\mathbf{z}; \mathbf{y})]^\top \\
&+ [\nabla \log p(\mathbf{z}; \mathbf{y})][\nabla \tilde{F}(\mathbf{y}; \mathbf{z})]^\top + \nabla^2 \tilde{F}(\mathbf{y}; \mathbf{z}) + \tilde{F}(\mathbf{y}; \mathbf{z}) \nabla^2 \log p(\mathbf{z}; \mathbf{y}).
\end{aligned}
\tag{11}
$$

*Then,* $\tilde{\nabla}^2 F(\mathbf{y}; \mathbf{z})$ *is an unbiased estimator of* $\nabla^2 F(\mathbf{y})$*, i.e.,* $\mathbb{E}_{\mathbf{z} \sim p(\mathbf{z}; \mathbf{y})}[\tilde{\nabla}^2 F(\mathbf{y}; \mathbf{z})] = \nabla^2 F(\mathbf{y})$.

The result in Lemma 1 shows how to evaluate an unbiased estimator of the Hessian $\nabla^2 F(\mathbf{y})$. If we consider $a$ as a random variable with a uniform distribution over the interval $[0, 1]$, then we can define the random variable $\mathbf{z}(a)$ with the probability distribution $p(\mathbf{z}(a); \mathbf{x}(a))$ where $\mathbf{x}(a)$ is defined as $\mathbf{x}(a) := a \cdot \mathbf{x}^t + (1 - a) \cdot \mathbf{x}^{t-1}$. Considering these two random variables and the result in Lemma 1, we can construct an unbiased estimator of the integral $\int_0^1 \nabla^2 F(\mathbf{x}(a)) \mathrm{d}a$ in (9) by

$$
\tilde{\nabla}_t^2 \overset{\text{def}}{=} \frac{1}{|\mathcal{M}|} \sum_{(a, \mathbf{z}(a)) \in \mathcal{M}} \tilde{\nabla}^2 F(\mathbf{x}(a); \mathbf{z}(a)),
\tag{12}
$$

where $\mathcal{M}$ is a minibatch containing $|\mathcal{M}|$ samples of random tuple $(a, \mathbf{z}(a))$. Once we have access to $\tilde{\nabla}_t^2$ which is an unbiased estimator of $\int_0^1 \nabla^2 F(\mathbf{x}(a)) \mathrm{d}a$, we can approximate the gradient difference $\Delta^t$ by its unbiased estimator which is defined as

$$
\tilde{\Delta}^t := \tilde{\nabla}_t^2 (\mathbf{x}^t - \mathbf{x}^{t-1}).
\tag{13}
$$

Note that for the general objective $F(\cdot)$, the matrix-vector product $\tilde{\nabla}_t^2 (\mathbf{x}^t - \mathbf{x}^{t-1})$ requires $\mathcal{O}(d^2)$ computation and memory. To resolve this issue, in Section 4.1 we provide an implementation of (13) using only first-order information which has a computational and memory complexity of $\mathcal{O}(d)$. Using $\tilde{\Delta}^t$ as an unbiased estimator of the gradient difference $\Delta^t$, we define our gradient estimator as

$$
\mathbf{g}^t = \nabla \tilde{F}(\mathbf{x}^0; \mathcal{M}_0) + \sum_{i=1}^t \tilde{\Delta}^t.
\tag{14}
$$

This update can also be written in a recursive way as $\mathbf{g}^t = \mathbf{g}^{t-1} + \tilde{\Delta}^t$, if we set $\mathbf{g}^0 = \nabla \tilde{F}(\mathbf{x}^0; \mathcal{M}_0)$. Note that the proposed approach for gradient approximation in (14) has a variance reduction mechanism which leads to optimal computational complexity of SCG++ in terms of number of calls to the stochastic oracle. We further highlight this point in Section 4.2.

## 4.1 Implementation of the Hessian-Vector Product

Now we focus on the computation of the gradient difference approximation $\tilde{\Delta}^t$ in (13). We aim to come up with a scheme that avoids explicitly computing the matrix estimator $\tilde{\nabla}_t^2$ which has a

complexity of $\mathcal{O}(d^2)$, and present an approach directly approximating $\tilde{\Delta}^t$ that only uses the finite differences of gradients with a complexity of $\mathcal{O}(d)$. Based on (12), computing $\tilde{\nabla}_t^2(\mathbf{x}^t - \mathbf{x}^{t-1})$ is equivalent to computing $|\mathcal{M}|$ instances of $\tilde{\nabla}^2 F(\mathbf{y};\mathbf{z})(\mathbf{x}^t - \mathbf{x}^{t-1})$ for some $\mathbf{y} \in \mathcal{C}$ and $\mathbf{z} \in \mathcal{Z}$. Denote $\mathbf{d} = \mathbf{x}^t - \mathbf{x}^{t-1}$ and use the expression in (11) to write

$$
\begin{aligned}
\tilde{\nabla}^2 F(\mathbf{y};\mathbf{z}) \cdot \mathbf{d} = {} & \tilde{F}(\mathbf{y};\mathbf{z})[\nabla \log p(\mathbf{z};\mathbf{y})^\top \mathbf{d}]\nabla \log p(\mathbf{z};\mathbf{y}) + [\nabla \log p(\mathbf{z};\mathbf{y})^\top \mathbf{d}]\nabla \tilde{F}(\mathbf{x};\mathbf{z}) \\
& + [\nabla \tilde{F}(\mathbf{y};\mathbf{z})^\top \mathbf{d}][\nabla \log p(\mathbf{z};\mathbf{y})] + \nabla^2 \tilde{F}(\mathbf{y};\mathbf{z}) \cdot \mathbf{d} + \tilde{F}(\mathbf{y};\mathbf{z})\nabla^2 \log p(\mathbf{z};\mathbf{y}) \cdot \mathbf{d}.
\end{aligned}
\tag{15}
$$

Note that the first three terms can be computed in time $\mathcal{O}(d)$ and only the last two terms on the right hand side of (15) involve $\mathcal{O}(d^2)$ operations, which can be approximated by the following finite gradient difference scheme. For any twice differentiable function $\psi : \mathbb{R}^d \to \mathbb{R}$ and arbitrary $\mathbf{d} \in \mathbb{R}^d$ with bounded Euclidean norm $\|\mathbf{d}\| \leq D$, we compute, for some small $\delta > 0$,

$$
\phi(\delta; \psi) \overset{\text{def}}{=} \frac{\nabla\psi(\mathbf{y} + \delta \cdot \mathbf{d}) - \nabla\psi(\mathbf{y} - \delta \cdot \mathbf{d})}{2\delta} \simeq \nabla^2 \psi(\mathbf{y}) \cdot \mathbf{d}.
\tag{16}
$$

As the Hessian of $\psi(\cdot)$ is $L_2$-smooth, the above approximation can be bounded by $\|\nabla^2\psi(\mathbf{y}) \cdot \mathbf{d} - \phi(\delta;\psi)\| = \|\nabla^2\psi(\mathbf{y})\cdot\mathbf{d} - \nabla^2\psi(\tilde{\mathbf{x}})\cdot\mathbf{d}\| \leq D^2 L_2 \delta$, where $\tilde{\mathbf{x}}$ is obtained from the mean value theorem. This quantity can be made arbitrary small by decreasing $\delta$. In next section, we show that setting $\delta = \mathcal{O}(\epsilon^2)$ is sufficient, where $\epsilon$ is the target accuracy. By applying the technique of (16) to the two functions $\psi(\mathbf{y}) = \tilde{F}(\mathbf{y};\mathbf{z})$ and $\psi(\mathbf{y}) = \log p(\mathbf{z};\mathbf{y})$, we can approximate (15) in time $\mathcal{O}(d)$:

$$
\begin{aligned}
\xi_\delta(\mathbf{y};\mathbf{z}) \overset{\text{def}}{=} {} & \tilde{F}(\mathbf{y};\mathbf{z})[\nabla \log p(\mathbf{z};\mathbf{y})^\top \mathbf{d}]\nabla \log p(\mathbf{z};\mathbf{y}) + [\nabla \log p(\mathbf{z};\mathbf{y})^\top \mathbf{d}]\nabla \tilde{F}(\mathbf{x};\mathbf{z}) \\
& + [\nabla \tilde{F}(\mathbf{y};\mathbf{z})^\top \mathbf{d}][\nabla \log p(\mathbf{z};\mathbf{y})] + \phi(\delta; \tilde{F}(\mathbf{y};\mathbf{z})) + \phi(\delta; \log p(\mathbf{z};\mathbf{y})).
\end{aligned}
\tag{17}
$$

We further can define a minibatch version of that which is used in Option II of Step 8 in Algorithm 1,

$$
\xi_\delta(\mathbf{x};\mathcal{M}) \overset{\text{def}}{=} \frac{1}{|\mathcal{M}|} \sum_{(a,\mathbf{z}(a)) \in \mathcal{M}} \xi_\delta(\mathbf{x}(a); \mathbf{z}(a)).
\tag{18}
$$

## 4.2 Convergence Analysis

In this section, we analyze the convergence of Algorithm 1 using (18) as the gradient-difference estimation. The result for (13) can be obtained similarly. We note that (13) is a special case of (18) by taking $\delta \to 0$ (e.g., by letting $\delta = O(\epsilon^2)$). We first state the assumptions required for our analysis.

**Assumption 4.1** (function value at the origin). *The function value $F$ at the origin is $F(\mathbf{0}) \geq 0$.*

**Assumption 4.2** (bounded stochastic function value). *The stochastic function $\tilde{F}(\mathbf{x};\mathbf{z})$ has bounded function value for all $\mathbf{z} \in \mathcal{Z}$ and $\mathbf{x} \in \mathcal{C}$: $\max_{\mathbf{z} \in \mathcal{Z}, \mathbf{x} \in \mathcal{C}} \tilde{F}(\mathbf{x};\mathbf{z}) \leq B$.*

**Assumption 4.3** (monotonicity and DR-submodularity). *$F$ is monotone and DR-submodular.*

**Assumption 4.4** (compactness of feasible domain). *The set $\mathcal{C}$ is compact with diameter $D$.*

**Assumption 4.5** (bounded gradient norm). *For all $\mathbf{x} \in \mathcal{C}$, the stochastic gradient $\nabla \tilde{F}$ has bounded norm: $\forall \mathbf{z} \in \mathcal{Z}, \|\nabla \tilde{F}(\mathbf{x};\mathbf{z})\| \leq G_{\tilde{F}}$, and the norm of the gradient of $\log p$ has bounded fourth-order moment, i.e., $\mathbb{E}_{\mathbf{z} \sim p(\mathbf{x};\mathbf{z})}\|\nabla \log p(\mathbf{z};\mathbf{x})\|^4 \leq G_p^4$. Further we define $G = \max\{G_{\tilde{F}}, G_p\}$.*

**Assumption 4.6** (bounded second-order derivatives). *$\forall \mathbf{x} \in \mathcal{C}$, the Hessian $\nabla^2 \tilde{F}$ has bounded spectral norm $\forall \mathbf{z} \in \mathcal{Z}, \|\nabla^2 \tilde{F}(\mathbf{x};\mathbf{z})\| \leq L_{\tilde{F}}$, and spectral norm of the log-probability Hessian has bounded second moment: $\mathbb{E}_{\mathbf{z} \sim p(\mathbf{z};\mathbf{x})}\|\nabla^2 \log p(\mathbf{z};\mathbf{x})\|^2 \leq L_p^2$. Further we define $L = \max\{L_{\tilde{F}}, L_p\}$.*

**Assumption 4.7** (continuity of the Hessian). *The stochastic Hessian is $L_{2,f}$-Lipschitz continuous, i.e, for all $\mathbf{x},\mathbf{y} \in \mathcal{C}$ and all $\mathbf{z} \in \mathcal{Z}$, i.e., $\|\nabla^2 \tilde{F}(\mathbf{x};\mathbf{z}) - \nabla^2 \tilde{F}(\mathbf{y};\mathbf{z})\| \leq L_{2,\tilde{F}}\|\mathbf{x} - \mathbf{y}\|$. The Hessian of the log probability $\log p(\mathbf{x};\mathbf{z})$ is $L_{2,p}$-Lipschitz continuous: for all $\mathbf{x},\mathbf{y} \in \mathcal{C}$ and all $\mathbf{z} \in \mathcal{Z}$, i.e., $\|\nabla^2 \log p(\mathbf{x};\mathbf{z}) - \nabla^2 \log p(\mathbf{y};\mathbf{z})\| \leq L_{2,p}\|\mathbf{x} - \mathbf{y}\|$. Further, define $L_2 = \max\{L_{2,\tilde{F}}, L_{2,p}\}$.*

**Remark 1.** *Assumption 4.7 is only used to show the finite difference scheme (15) has bounded variance, and the oracle complexity of our method does not depend on $L_{2,\tilde{F}}$ and $L_{2,p}$.*

As we mentioned in the previous section, the update for the stochastic gradient vector $\mathbf{g}^t$ in the update of SCG++ is designed properly to reduce the noise of gradient approximation. In the following lemma, we formally characterize the variance of gradient approximation for SCG++ . To this end, we also need to properly choose the minibatch sizes $|\mathcal{M}_0|$ and $|\mathcal{M}|$.

**Lemma 2.** *Consider SCG++ outlined in Algorithm 1 and assume that in Step 8 we follow* (18) *to construct the gradient difference approximation $\tilde{\Delta}^t$ (Option II). If Assumptions (4.2), (4.4), (4.5), (4.6), and (4.7) hold and we set the minibatch sizes to $|\mathcal{M}_0| = (G^2/(\bar{L}^2 D^2 \epsilon^2))$ and $|\mathcal{M}| = 2/\epsilon$, and the error of Hessian-vector product approximation $\delta$ is $\mathcal{O}(\epsilon^2)$ as in (31), then*

$$\mathbb{E}\left[\|\mathbf{g}^t - \nabla F(\mathbf{x}^t)\|^2\right] \leq (1 + \epsilon t)\bar{L}^2 D^2 \epsilon^2, \quad \forall t \in \{0, \dots, T-1\}, \tag{19}$$

*where $\bar{L}$ is a constant defined by $\bar{L}^2 \stackrel{\text{def}}{=} 4B^2 G^4 + 16G^4 + 4L^2 + 4B^2 L^2$.*

Lemma 2 shows that by $|\mathcal{M}| = \mathcal{O}(\epsilon^{-1})$ calls to the stochastic oracle at each iteration, the variance of gradient approximation in SCG++ after $t$ iterations is of order $\mathcal{O}((1 + \epsilon t)\epsilon)$. In the following theorem, we incorporate this result to characterize the convergence guarantee of SCG++ .

**Theorem 1.** *Consider the SCG++ method outlined in Algorithm 1 and assume that in Step 8 we follow the update in* (18) *to construct the gradient difference approximation $\tilde{\Delta}^t$ (Option II). If Assumptions 4.1-4.7 hold, then the output of SCG++ denoted by $\mathbf{x}^T$ satisfies*

$$\mathbb{E}\left[F(\mathbf{x}^T)\right] \geq (1 - 1/e)F(\mathbf{x}^*) - 2\bar{L}D^2\epsilon,$$

*by setting $|\mathcal{M}_0| = \frac{G^2}{2\bar{L}^2 D^2 \epsilon^2}$, $|\mathcal{M}| = \frac{1}{2\epsilon}$, $T = \frac{1}{\epsilon}$, and $\delta = \mathcal{O}(\epsilon^2)$ as in (31). Here $\bar{L}$ is a constant defined by $\bar{L}^2 \stackrel{\text{def}}{=} 4B^2 G^4 + 16G^4 + 4L^2 + 4B^2 L^2$.*

The result in Theorem 1 shows that after at most $T = \mathcal{O}(1/\epsilon)$ iterations the objective function value for the output of SCG++ is at least $(1 - 1/e)\text{OPT} - \mathcal{O}(\epsilon)$. As the number of calls to the stochastic oracle per iteration is of $\mathcal{O}(1/\epsilon)$, to reach a $[(1 - 1/e)\text{OPT} - \mathcal{O}(\epsilon)]$ approximation guarantee the SCG++ method has an overall stochastic first-order oracle complexity of $\mathcal{O}(1/\epsilon^2)$. We formally characterize this result in the following corollary.

**Corollary 1** (oracle complexities). *To find a $[(1 - 1/e)\text{OPT} - \epsilon]$ solution to Problem (1) using Algorithm 1 with Option II, the overall stochastic first-order oracle complexity is $(2G^2 D^2 + 4\bar{L}^2 D^4)/\epsilon^2$ and the overall linear optimization oracle complexity is $2\bar{L}D^2/\epsilon$.*

## 5 Discrete Stochastic Submodular Maximization

In this section, we focus on extending our result in the previous section to the case where $F$ is the multilinear extension of a (stochastic) discrete submodular function $f$. This is also an instance of the non-oblivious stochastic optimization in (1). Indeed, once such a result is achieved, with proper rounding scheme such as randomized pipage rounding [6] or contention resolution method [32], we can extend our results to the discrete setting. Let $V$ denote a finite set of $d$ elements, i.e., $V = \{1, \dots, d\}$. Consider a discrete submodular function $f : 2^V \to \mathbb{R}_+$, which is defined as an *expectation* over a set of functions $f_\gamma : 2^V \to \mathbb{R}_+$. Our goal is to maximize $f$ subject to some constraint $\mathcal{I}$, where $\mathcal{I}$ contains feasible subsets of $V$. In other words, we aim to solve the following discrete and stochastic submodular function maximization problem

$$\max_{S \in \mathcal{I}} f(S) := \max_{S \in \mathcal{I}} \mathbb{E}_{\gamma \sim p(\gamma)}[f_\gamma(S)], \tag{20}$$

where $p(\gamma)$ is an arbitrary distribution. In particular, we assume the pair $M = \{V, \mathcal{I}\}$ forms a matroid with rank $r$. The prototypical example is maximization under the cardinality constraint, i.e., for a given integer $r$, find $S \subseteq V, |S| \leq r$, which maximizes $f$. The challenge here is to find a solution with near-optimal quality for the problem in (20) without computing the expectation in (20). That is, we assume access to an oracle that, given a set $S$, outputs an independently chosen sample $f_\gamma(S)$ where $\gamma \sim p(\gamma)$. The focus of this section is on extending our result into the discrete domain and showing that SCG++ can be applied for maximizing a stochastic submodular set function $f$, namely Problem (20), through the multilinear extension of the function $f$. Specifically, in lieu of solving (20) we can solve its multilinear extension problem

$$\max_{\mathbf{x} \in \mathcal{C}} F(\mathbf{x}), \tag{21}$$

where $F : [0,1]^V \to \mathbb{R}_+$ is the multilinear extension of $f$ and is defined as

$$F(\mathbf{x}) := \sum_{S \subseteq V} f(S) \prod_{i \in S} x_i \prod_{j \notin S} (1 - x_j) = \sum_{S \subseteq V} \mathbb{E}_{\gamma \sim p(\gamma)}[f_\gamma(S)] \prod_{i \in S} x_i \prod_{j \notin S} (1 - x_j), \tag{22}$$

and the convex set $\mathcal{C} = \text{conv}\{1_I : I \in \mathcal{I}\}$ is the matroid polytope [6]. Note that here $x_i$ denotes the $i$-th component of the vector $\mathbf{x}$. In other words, $F(\mathbf{x})$ is the expected value of $f$ over sets wherein each element $i$ is included with probability $x_i$ independently. To solve (21) using SCG++ , we need access to unbiased estimators of the gradient and the Hessian. We now construct the Hessian approximation $\tilde{\nabla}_k^2$ using the result in [6] which is stated in Lemma 4, in the supplementary material.

Let $a$ be a uniform random variable between $[0,1]$ and let $\mathbf{e} = (e_1, \cdots, e_d)$ be a random vector in which $e_i$'s are generated i.i.d. according to the uniform distribution over the unit interval $[0,1]$. In each iteration, a minibatch $\mathcal{M}$ of $|\mathcal{M}|$ samples of $\{a, \mathbf{e}, \gamma\}$ (recall that $\gamma$ is the random variable that parameterizes the component function $f_\gamma$), i.e. $\mathcal{M} = \{a_k, \mathbf{e}_k, \gamma_k\}_{k=1}^{|\mathcal{M}|}$, is generated. Then for all $k \in [|\mathcal{M}|]$, we let $\mathbf{x}_{a_k} = a_k \mathbf{x}^t + (1 - a_k)\mathbf{x}^{t-1}$ and construct the random set $S(\mathbf{x}_{a_k}, \mathbf{e}_k)$ using $\mathbf{x}_{a_k}$ and $\mathbf{e}_k$ in the following way: $s \in S(\mathbf{x}_{a_k}, \mathbf{e}_k)$ if and only if $[\mathbf{e}_k]_s \leq [\mathbf{x}_{a_k}]_s$ for $s \in [d]$. Having $S(\mathbf{x}_{a_k}, \mathbf{e}_k)$ and $\gamma_k$, each entry of the Hessian estimator $\tilde{\nabla}_t^2 \in \mathbb{R}^{d \times d}$ is

$$
\begin{aligned}
\left[\tilde{\nabla}_t^2\right]_{i,j} = \frac{1}{|\mathcal{M}|} \sum_{k \in [|\mathcal{M}|]} & f_{\gamma_k}(S(\mathbf{x}_{a_k}, \mathbf{e}_k) \cup \{i, j\}) - f_{\gamma_k}(S(\mathbf{x}_{a_k}, \mathbf{e}_k) \cup \{i\} \setminus \{j\}) \\
& - f_{\gamma_k}(S(\mathbf{x}_{a_k}, \mathbf{e}_k) \cup \{j\} \setminus \{i\}) + f_{\gamma_k}(S(\mathbf{x}_{a_k}, \mathbf{e}_k) \setminus \{i, j\}),
\end{aligned}
\tag{23}
$$

where $i \neq j$, and if $i = j$ then $[\tilde{\nabla}_t^2]_{i,j} = 0$. As linear optimization over the rank-$r$ matroid polytope always return $\mathbf{v}^t$ with at most $r$ nonzero entries, the complexity of computing (23) is $\mathcal{O}(|\mathcal{M}|rd)$.

We use the above Hessian approximation to solve (21) as a special case of Problem (1) using SCG++ .

**Theorem 2.** *Consider $D_\gamma := \max_{i \in V} f_\gamma(i)$ as the maximum marginal value of $f_\gamma$, and define $D_f := \sqrt{\mathbb{E}_\gamma[D_\gamma^2]}$. By using the minibatch size $|\mathcal{M}| = \mathcal{O}(\sqrt{r^3 d} D_f / \epsilon)$ and $|\mathcal{M}_0| = \mathcal{O}(\sqrt{d} D_f / \sqrt{r} \epsilon^2)$, Algorithm 1 finds a $[(1 - 1/e)OPT - 6\epsilon]$ approximation of the multilinear extension problem in (21) at most $(\sqrt{r^3 d} D_f / \epsilon)$ iterations. Moreover, the overall stochastic oracle cost is $\mathcal{O}(r^3 d D_f^2 / \epsilon^2)$.*

Since the cost of a single stochastic gradient computation is $\mathcal{O}(d)$, Theorem 2 shows that the overall computation complexity of Algorithm 1 is $\mathcal{O}(d^2/\epsilon^2)$. Note that, in multilinear extension case, the smoothness Assumption 4.6 required for the results in Section 4 is absent, and that is why we need to develop a more sophisticated gradient-difference estimator to achieve a similar theoretical guarantee (more details is available in the appendix).

**Remark 2** (optimality of oracle complexities). *Note that to achieve the tight $(1 - 1/e - \epsilon)$ approximation, the $\mathcal{O}(1/\epsilon^2)$ stochastic oracle complexity in Theorem 2 is optimal in terms of its dependency on $\epsilon$. A lower bound on the stochastic oracle complexity is given in the following theorem.*

# 6 Lower Bound

In this section, we show that reaching a $(1 - 1/e - \epsilon)$-optimal solution of Problem (1) requires at least $\mathcal{O}(1/\epsilon^2)$ calls to an oracle which provides stochastic first-order information. To do so, we first construct a stochastic submodular set function $f$, defined through the expectation $f(S) = \mathbb{E}_{\gamma \sim p(\gamma)}[f_\gamma(S)]$, with the following property: Obtaining a $(1 - 1/e - \epsilon)$-optimal solution for maximization of $f$ under a cardinality constraint (an instance of Problem (20)) requires at least $\mathcal{O}(1/\epsilon^2)$ samples of the form $f_\gamma(\cdot)$ where $\gamma$ is generated i.i.d from distribution $p$. Such a lower bound on sample complexity can be directly extended to Problem (1) with an stochastic first order oracle, by considering the multilinear extension of the function $f$, denoted by $F$, and noting that (i) Problems (20) and (21) have the same optimal values, and (ii) one can construct an unbiased estimator of the gradient of the multilinear extension using $d$ independent samples from the underlying stochastic set function $f$. Hence, any method for maximizing (21) is also an algorithm for maximizing (20) with the same guarantees on the quality of the solution and with sample complexities that differ at most by a factor $d$. Now we provide the formal statements regarding the above argument.

**Theorem 3.** *There exists a distribution $p(\gamma)$ and a monotone submodular function $f : 2^V \to \mathbb{R}_+$, given as $f(S) = \mathbb{E}_{\gamma \sim p(\gamma)}[f_\gamma(S)]$, such that the following holds: In order to find a $(1 - 1/e - \epsilon)$-optimal solution for (20) with $k$-cardinality constraint, any algorithm requires at least $\min\{\exp(\alpha k), \beta/\epsilon^2\}$ stochastic samples $f_\gamma(\cdot)$.*

**Corollary 2.** *There exists a DR-submodular function $F : [0,1]^n \to \mathbb{R}$, a convex constraint $\mathcal{C}$, and a stochastic first order oracle $\mathcal{O}_{first}$, such that any algorithm for maximizing $F$ subject to $\mathcal{C}$ requires at least $\min\{\exp(\alpha n), \beta/\epsilon^2\}$ queries from $\mathcal{O}_{first}$.*

## 7  Conclusion

In this paper, we developed SCG++ , the first efficient variant of continuous greedy for maximizing a stochastic continuous DR-submodular function subject to a convex constraint. We showed that SCG++ achieves a tight $[(1 - 1/e)\text{OPT} - \epsilon]$ solution while using $O(1/\epsilon^2)$ stochastic gradients. We further derived a tight lower bound on the number of calls to the first-order stochastic oracle for achieving a $[(1 - 1/e)\text{OPT} - \epsilon]$ approximate solution. This result showed that SCG++ has the optimal sample complexity for finding an optimal $(1 - 1/e)$ approximation guarantee for monotone but stochastic DR-submodular functions.

## Acknowledgment

The work of H. Hassani was partially supported by NSF CPS-1837253. Karbasi's work is partially supported by NSF (IIS-1845032), ONR (N00014- 19-1-2406) and AFOSR (FA9550-18-1-0160). Shen's work is supported by Zhejiang Provincial Natural Science Foundation of China under Grant No. LZ18F020002, and National Natural Science Foundation of China (Grant No: 61672376, 61751209, 61472347).

## Footnotes

*The authors are listed in alphabetical order.

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
