[Supplementary Material]

# 8 Supplementary Material

## 8.1 Proof of Lemma 1

*Proof.* Recall the definition of $F(\mathbf{y}) = \int_{\mathbf{z} \in \mathcal{Z}} \tilde{F}(\mathbf{y}; \mathbf{z}) p(\mathbf{z}; \mathbf{y}) d\mathbf{z}$. The first order differential can be computed by

$$
\begin{aligned}
\nabla F(\mathbf{y}) &= \int_{\mathbf{z} \in \mathcal{Z}} p(\mathbf{z}; \mathbf{y}) \nabla \tilde{F}(\mathbf{y}; \mathbf{z}) + \tilde{F}(\mathbf{y}; \mathbf{z}) \nabla p(\mathbf{z}; \mathbf{y}) d\mathbf{z} \\
&= \int_{\mathbf{z} \in \mathcal{Z}} p(\mathbf{z}; \mathbf{y}) \left[ \nabla \tilde{F}(\mathbf{y}; \mathbf{z}) + \tilde{F}(\mathbf{y}; \mathbf{z}) \nabla \log p(\mathbf{z}; \mathbf{y}) \right] d\mathbf{z},
\end{aligned}
\tag{24}
$$

where we use $\nabla \log p(\mathbf{z}; \mathbf{y}) = \frac{\nabla p(\mathbf{z}; \mathbf{y})}{p(\mathbf{z}; \mathbf{y})}$ in the second equality. We now compute the second order differential by

$$
\begin{aligned}
\nabla^2 F(\mathbf{y}) &= \int_{\mathbf{z} \in \mathcal{Z}} \left[ \nabla \tilde{F}(\mathbf{y}; \mathbf{z}) + \tilde{F}(\mathbf{y}; \mathbf{z}) \nabla \log p(\mathbf{z}; \mathbf{y}) \right] [\nabla p(\mathbf{z}; \mathbf{y})]^\top d\mathbf{z} \\
&\quad + \int_{\mathbf{z} \in \mathcal{Z}} p(\mathbf{z}; \mathbf{y}) \left[ \nabla^2 \tilde{F}(\mathbf{y}; \mathbf{z}) + [\nabla \log p(\mathbf{z}; \mathbf{y})][\nabla \tilde{F}(\mathbf{y}; \mathbf{z})]^\top + \tilde{F}(\mathbf{y}; \mathbf{z}) \nabla^2 \log p(\mathbf{z}; \mathbf{y}) \right] d\mathbf{z} \\
&= \int_{\mathbf{z} \in \mathcal{Z}} p(\mathbf{z}; \mathbf{y}) \left[ \nabla \tilde{F}(\mathbf{y}; \mathbf{z}) + \tilde{F}(\mathbf{y}; \mathbf{z}) \nabla \log p(\mathbf{z}; \mathbf{y}) \right] [\nabla \log p(\mathbf{z}; \mathbf{y})]^\top d\mathbf{z} \\
&\quad + \int_{\mathbf{z} \in \mathcal{Z}} p(\mathbf{z}; \mathbf{y}) \left[ \nabla^2 \tilde{F}(\mathbf{y}; \mathbf{z}) + [\nabla \log p(\mathbf{z}; \mathbf{y})][\nabla \tilde{F}(\mathbf{y}; \mathbf{z})]^\top + \tilde{F}(\mathbf{y}; \mathbf{z}) \nabla^2 \log p(\mathbf{z}; \mathbf{y}) \right] d\mathbf{z},
\end{aligned}
$$

where again we use $\nabla \log p(\mathbf{z}; \mathbf{y}) = \frac{\nabla p(\mathbf{z}; \mathbf{y})}{p(\mathbf{z}; \mathbf{y})}$ in the second equality. From such derivation, we have the result. $\qquad\square$

## 8.2 Proof of Lemma 2

Before we give the proof of Lemma 2, we first present a lemma which bounds the second moment of the spectral norm of the Hessian estimator $\tilde{\nabla}^2 F(\mathbf{y}; \mathbf{z})$ for any $\mathbf{y} \in \mathcal{C}$.

**Lemma 3.** *Recall the definition of the Hessian estimator $\tilde{\nabla}^2 F(\mathbf{y}; \mathbf{z})$ in (11). Under Assumption 4.2, 4.5, 4.6, we bound for any $\mathbf{y} \in \mathcal{C}$*

$$
\mathbb{E}_{\mathbf{z} \sim p(\mathbf{z}; \mathbf{y})} \|\tilde{\nabla}^2 F(\mathbf{y}; \mathbf{z})\|^2 \leq 4B^2 G^4 + 16G^4 + 4L^2 + 4B^2 L^2 \overset{\text{def}}{=} \bar{L}^2.
\tag{25}
$$

*Lemma 3.* From the definition of the Hessian estimator $\tilde{\nabla}^2 F(\mathbf{y}; \mathbf{z})$ (see (11)), we have

$$
\|\tilde{\nabla}^2 F(\mathbf{y}; \mathbf{z})\| \leq B\|\nabla \log p(\mathbf{z}; \mathbf{y})\|^2 + 2G\|\nabla \log p(\mathbf{z}; \mathbf{y})\| + L + B\|\nabla^2 \log p(\mathbf{z}; \mathbf{y})\|,
\tag{26}
$$

where we use Assumption 4.2 and 4.5 and the triangle inequality. Futher, taking expectation on both sides and use Assumption 4.6 to bound

$$
\mathbb{E}\|\tilde{\nabla}^2 F(\mathbf{y}; \mathbf{z})\|^2 \leq 4B^2 G^4 + 16G^4 + 4L^2 + 4B^2 L^2.
\tag{27}
$$

$\qquad\square$

*Lemma 2.* We prove via induction. When $t = 0$, use the unbiasedness of $\nabla \tilde{F}(\mathbf{x}^0; \mathbf{z})$ and Assumption 4.5, we bound

$$
\mathbb{E}_{\mathcal{M}_0}\|F(\mathbf{x}^0) - \mathbf{g}^0\|^2 = \frac{1}{|\mathcal{M}_0|} \mathbb{E}\|F(\mathbf{x}^0) - \nabla \tilde{F}(\mathbf{x}^0; \mathbf{z})\|^2 \leq \frac{1}{|\mathcal{M}_0|} \mathbb{E}\|\nabla \tilde{F}(\mathbf{x}^0; \mathbf{z})\|^2 \leq \frac{G^2}{|\mathcal{M}_0|} \leq \bar{L}^2 D^2 \epsilon^2.
$$

Now assume that we have the result for $t = \bar{t}$. When $t = \bar{t} + 1$, we have from the definition of $\mathbf{g}^t$

$$
\begin{aligned}
\mathbf{g}^t - \nabla F(\mathbf{x}^t) &= [\mathbf{g}^{t-1} - \nabla F(\mathbf{x}^{t-1})] + \left[ \xi_\delta(\mathbf{x}; \mathcal{M}) - \tilde{\nabla}_t^2(\mathbf{x}^t - \mathbf{x}^{t-1}) \right] \\
&\quad + \left[ \tilde{\nabla}_t^2(\mathbf{x}^t - \mathbf{x}^{t-1}) - (\nabla F(\mathbf{x}^t) - \nabla F(\mathbf{x}^{t-1})) \right].
\end{aligned}
$$

Expand $\|\nabla F(\mathbf{x}^t) - \mathbf{g}^t\|^2$ to obtain

$$\|\nabla F(\mathbf{x}^t) - \mathbf{g}^t\|^2 = \|\nabla F(\mathbf{x}^t) - \nabla F(\mathbf{x}^{t-1}) - \tilde{\nabla}_t^2(\mathbf{x}^t - \mathbf{x}^{t-1})\|^2 + \|\mathbf{g}^{t-1} - \nabla F(\mathbf{x}^{t-1})\|^2$$
$$+ 2\langle \nabla F(\mathbf{x}^t) - \nabla F(\mathbf{x}^{t-1}) - \tilde{\nabla}_t^2(\mathbf{x}^t - \mathbf{x}^{t-1}), \mathbf{g}^{t-1} - \nabla F(\mathbf{x}^{t-1})\rangle$$
$$+ 2\langle \tilde{\nabla}_t^2(\mathbf{x}^t - \mathbf{x}^{t-1}) - \xi_\delta(\mathbf{x};\mathcal{M}), \nabla F(\mathbf{x}^t) - \nabla F(\mathbf{x}^{t-1}) - \tilde{\nabla}_t^2(\mathbf{x}^t - \mathbf{x}^{t-1})\rangle$$
$$+ 2\langle \tilde{\nabla}_t^2(\mathbf{x}^t - \mathbf{x}^{t-1}) - \xi_\delta(\mathbf{x};\mathcal{M}), \mathbf{g}^{t-1} - \nabla F(\mathbf{x}^{t-1})\rangle$$
$$+ \|\tilde{\nabla}_t^2(\mathbf{x}^t - \mathbf{x}^{t-1}) - \xi_\delta(\mathbf{x};\mathcal{M})\|^2. \tag{28}$$

Using the unbiasedness of $\tilde{\nabla}_t^2(\mathbf{x}^t - \mathbf{x}^{t-1})$, we have

$$\mathbb{E}\langle \nabla F(\mathbf{x}^t) - \nabla F(\mathbf{x}^{t-1}) - \tilde{\nabla}_t^2(\mathbf{x}^t - \mathbf{x}^{t-1}), \mathbf{g}^{t-1} - \nabla F(\mathbf{x}^{t-1})\rangle = 0. \tag{29}$$

Additionally, from the unbiasedness of $\tilde{\Delta}^t$, we have

$$\mathbb{E}\|\tilde{\Delta}^t - (\nabla F(\mathbf{x}^t) - \nabla F(\mathbf{x}^{t-1}))\|^2 \le \frac{\epsilon^2 D^2}{|\mathcal{M}|}\mathbb{E}\|\tilde{\nabla}^2 F(\mathbf{x}(a_1); \mathbf{z}_1(a_1))\|^2 \le \frac{\epsilon^2 \bar{L}^2 D^2}{|\mathcal{M}|}, \tag{30}$$

where we use Lemma 3 in the last inequality. Taking expectation on both sides of (28), we have

$$\mathbb{E}\|\nabla F(\mathbf{x}^t) - \mathbf{g}^t\|^2$$
$$=\mathbb{E}\|\nabla F(\mathbf{x}^t) - \nabla F(\mathbf{x}^{t-1}) - \tilde{\nabla}_t^2(\mathbf{x}^t - \mathbf{x}^{t-1})\|^2 + \mathbb{E}\|\mathbf{g}^{t-1} - \nabla F(\mathbf{x}^{t-1})\|^2$$
$$+ 2\mathbb{E}\|\tilde{\nabla}_t^2(\mathbf{x}^t - \mathbf{x}^{t-1}) - \xi_\delta(\mathbf{x};\mathcal{M})\|\|\nabla F(\mathbf{x}^t) - \nabla F(\mathbf{x}^{t-1}) - \tilde{\nabla}_t^2(\mathbf{x}^t - \mathbf{x}^{t-1})\|$$
$$+ 2\mathbb{E}\|\tilde{\nabla}_t^2(\mathbf{x}^t - \mathbf{x}^{t-1}) - \xi_\delta(\mathbf{x};\mathcal{M})\|\|\mathbf{g}^{t-1} - \nabla F(\mathbf{x}^{t-1})\| + \mathbb{E}\|\tilde{\nabla}_t^2(\mathbf{x}^t - \mathbf{x}^{t-1}) - \xi_\delta(\mathbf{x};\mathcal{M})\|^2$$
$$\le\mathbb{E}\|\nabla F(\mathbf{x}^t) - \nabla F(\mathbf{x}^{t-1}) - \tilde{\nabla}_t^2(\mathbf{x}^t - \mathbf{x}^{t-1})\|^2 + \mathbb{E}\|\mathbf{g}^{t-1} - \nabla F(\mathbf{x}^{t-1})\|^2 + 4D^4 L_2^2\delta^2$$
$$+ 4D^2 L_2\delta\|\nabla F(\mathbf{x}^t) - \nabla F(\mathbf{x}^{t-1}) - \tilde{\nabla}_t^2(\mathbf{x}^t - \mathbf{x}^{t-1})\| + 4D^2 L_2\delta\|\mathbf{g}^{t-1} - \nabla F(\mathbf{x}^{t-1})\|$$
$$\le\frac{\bar{L}^2 D^2\epsilon^2}{|\mathcal{M}|} + (1 + \epsilon(t-1))\bar{L}^2 D^2\epsilon^2 + 4\delta\left(\frac{D^2 L_2\bar{L}D\epsilon}{\sqrt{|\mathcal{M}|}} + D^2 L_2\sqrt{(1+\epsilon(t-1))}\bar{L}D\epsilon + D^4 L_2^2\delta\right)$$

By taking $\delta$ sufficiently small such that

$$4\delta\left(\frac{D^2 L_2\bar{L}D\epsilon}{\sqrt{|\mathcal{M}|}} + D^2 L_2\sqrt{(1+\epsilon(t-1))}\bar{L}D\epsilon + D^4 L_2^2\delta\right) \le \bar{L}^2 D^2\epsilon^3/2, \tag{31}$$

we have shown that the induction holds for $t = \bar{t} + 1$. $\qquad\square$

## 8.3  Proof of Theorem 1

*Proof.* From Lemma 3, we have

$$\|\nabla^2 F(\mathbf{x})\|^2 \le \|\mathbb{E}_{\mathbf{z}\sim p(\mathbf{z};\mathbf{x})}\tilde{\nabla}^2 F(\mathbf{x};\mathbf{z})\|^2 \le \mathbb{E}_{\mathbf{z}\sim p(\mathbf{z};\mathbf{x})}\|\tilde{\nabla}^2 F(\mathbf{x};\mathbf{z})\|^2 \le \bar{L}^2. \tag{32}$$

From standard argument, $F$ can be proved to be $\bar{L}$-smooth. Let $\mathbf{x}^*$ be the global maximizer within the constraint set $\mathcal{C}$. From the smoothness of $F$, we have

$$F(\mathbf{x}^{t+1}) \ge F(\mathbf{x}^t) + \langle \nabla F(\mathbf{x}^t), \mathbf{x}^{t+1} - \mathbf{x}^t\rangle - \frac{\bar{L}}{2}\|\mathbf{x}^{t+1} - \mathbf{x}^t\|^2$$
$$= F(\mathbf{x}^t) + \frac{1}{T}\langle \nabla F(\mathbf{x}^t), \mathbf{v}^t\rangle - \frac{\bar{L}}{2T^2}\|\mathbf{v}^t\|^2 \tag{33}$$
$$= F(\mathbf{x}^t) + \frac{1}{T}\langle \mathbf{g}^t, \mathbf{v}^t\rangle + \frac{1}{T}\langle \nabla F(\mathbf{x}^t) - \mathbf{g}^t, \mathbf{v}^t\rangle - \frac{\bar{L}D^2}{2T^2}$$
$$\ge F(\mathbf{x}^t) + \frac{1}{T}\langle \mathbf{g}^t, \mathbf{x}^*\rangle + \frac{1}{T}\langle \nabla F(\mathbf{x}^t) - \mathbf{g}^t, \mathbf{v}^t\rangle - \frac{\bar{L}D^2}{2T^2},$$

where we use the optimality and boundedness of $\mathbf{v}^t$ in the last inequality. Take expectation on both sides and use the unbiasedness of $\mathbf{g}^t$ to yield

$$\mathbb{E}F(\mathbf{x}^{t+1}) \ge \mathbb{E}F(\mathbf{x}^t) + \frac{1}{T}\mathbb{E}\langle \nabla F(\mathbf{x}^t), \mathbf{x}^*\rangle + \frac{1}{T}\mathbb{E}\langle \nabla F(\mathbf{x}^t) - \mathbf{g}^t, \mathbf{v}^t\rangle - \frac{\bar{L}D^2}{2T^2}. \tag{34}$$

From the monotonicity of $F$ and the concavity of $F$ along positive directions, we have $\langle \nabla F(\mathbf{x}^t), \mathbf{x}^* \rangle \geq F(\mathbf{x}^*) - F(\mathbf{x}^t)$. Additionally, using the Young's inequality, we write

$$\mathbb{E}F(\mathbf{x}^{t+1}) \geq \mathbb{E}F(\mathbf{x}^t) + \frac{1}{T}\mathbb{E}[F(\mathbf{x}^*) - F(\mathbf{x}^t)] - \frac{1}{2\bar{L}}\mathbb{E}\|\nabla F(\mathbf{x}^t) - \mathbf{g}^t\|^2 - \frac{\bar{L}D^2}{T^2}.$$

Using Lemma 2, we have for all $t \in \{0, \ldots, T-1\}$

$$\mathbb{E}\|\nabla F(\mathbf{x}^t) - \mathbf{g}^t\|^2 \leq 2\bar{L}^2 D^2 \epsilon^2. \tag{35}$$

Consequently, we have with $T = \frac{1}{\epsilon}$

$$\mathbb{E}F(\mathbf{x}^{t+1}) \geq \mathbb{E}F(\mathbf{x}^t) + \epsilon\mathbb{E}[F(\mathbf{x}^*) - F(\mathbf{x}^t)] - 2\bar{L}\epsilon^2 D^2,$$

which is equivalent to

$$\mathbb{E}[F(\mathbf{x}^*) - F(\mathbf{x}^{t+1})] \leq (1-\epsilon)^T \mathbb{E}[F(\mathbf{x}^*) - F(\mathbf{x}^t)] - 2\bar{L}\epsilon D^2.$$

In conclusion, we have

$$\mathbb{E}F(\mathbf{x}^T) \geq (1 - 1/e)\mathbb{E}[F(\mathbf{x}^*)] - 2\bar{L}\epsilon D^2.$$

$\square$

## 8.4 Multilinear Extension as Non-oblivious Stochastic Optimization

We proceed to show that the problem in (22) is captured by (1). To do so, use $\text{Ber}(b; m)$ with $b \in \{0, 1\}$ and $m \in [0, 1]$ to denote the Bernoulli distribution with parameter $m$, i.e.

$$\text{Ber}(b; m) = m^b(1-m)^{1-b}.$$

Define the underlying distribution $p(\mathbf{z}, \gamma; \mathbf{x})$ as

$$p(\mathbf{z}, \gamma; \mathbf{x}) = p(\gamma) \times \prod_{i=1}^{d} \text{Ber}(\mathbf{z}_i; \mathbf{x}_i), \tag{36}$$

where $p(\gamma)$ is defined in (20), $\mathbf{z}_i$ is the $i^{th}$ entry of $\mathbf{z}$, and $\mathbf{x}_i$ is the $i^{th}$ entry of $\mathbf{x}$. Let $N(\mathbf{z})$ be a subset of $N$ such that $i \in N(\mathbf{z})$ iff $\mathbf{z}_i = 1$. We then define the stochastic function $\tilde{F}(\mathbf{x}; \mathbf{z}, \gamma)$ as

$$\tilde{F}(\mathbf{x}; \mathbf{z}, \gamma) = f_\gamma(N(\mathbf{z})), \tag{37}$$

where $f_\gamma$ is defined in (20). We emphasize that for a fixed $\mathbf{z}$ the stochastic function $\tilde{F}$ does not depend on $\mathbf{x}$ and hence $\nabla\tilde{F}(\mathbf{x}; \mathbf{z}) = 0$. Considering the definition of the stochastic function $\tilde{F}(\mathbf{x}; \mathbf{z}, \gamma)$ in (37), the multilinear extension function $F$ in (22), and the probability distribution $p(\mathbf{z}, \gamma; \mathbf{x})$ in (36) it can be verified that $F$ is the expectation of the random $\tilde{F}(\mathbf{x}; \mathbf{z}, \gamma)$, and, therefore, the problem in (22) can be written as (1).

At the first glance, it seems that we can apply the SCG++ method to maximize the multilinear extension function $F$. However, the smoothness conditions required for the result in Theorem 1 do not hold in the multilinear setting. To be more specific, following the result in Lemma 1, we can derive an unbiased estimator for the second-order differential of (22) using

$$\tilde{\nabla}^2 F(\mathbf{y}; \mathbf{z}) = \tilde{F}(\mathbf{y}; \mathbf{z}) \left[ [\nabla \log p(\mathbf{z}, \gamma; \mathbf{y})][\nabla \log p(\mathbf{z}, \gamma; \mathbf{y})]^\top + \nabla^2 \log p(\mathbf{z}, \gamma; \mathbf{y}) \right],$$

$$= f_\gamma(N(\mathbf{z})) \left[ [\sum_{i=1}^{d} \nabla \log \text{Ber}(\mathbf{z}_i; \mathbf{x}_i)][\sum_{i=1}^{d} \nabla \log \text{Ber}(\mathbf{z}_i; \mathbf{x}_i)]^\top + \sum_{i=1}^{d} \nabla^2 \log \text{Ber}(\mathbf{z}_i; \mathbf{x}_i) \right], \tag{38}$$

where we use $\nabla\tilde{F}(\mathbf{x}; \mathbf{z}) = 0$ in the first equality and use (36) and (37) in the second one. Further, note that $[\nabla \log \text{Ber}(\mathbf{z}_i; \mathbf{x}_i)]^2 + \nabla^2 \log \text{Ber}(\mathbf{z}_i; \mathbf{x}_i) = 0$ for all $i \in [d]$ and hence, the above estimator can be further simplified to

$$\tilde{\nabla}^2 F(\mathbf{y}; \mathbf{z}, \gamma) = f_\gamma(N(\mathbf{z})) \sum_{i,j=1}^{d} \mathbb{1}_{i \neq j} [\nabla \log \text{Ber}(\mathbf{z}_i; \mathbf{x}_i)][\nabla \log \text{Ber}(\mathbf{z}_j; \mathbf{x}_j)]^\top. \tag{39}$$

Despite the simple form of (39), the smoothness property in Assumption 4.6 is absent since every entry in the matrix $\tilde{\nabla}^2 F(\mathbf{y}; \mathbf{z}, \gamma)$ can have unbounded second-order moment when $\mathbf{x}_i \to 0$ or $\mathbf{x}_i \to 1$.

---

**Algorithm 2** (SCG++ ) for Multilinear Extension

---

**Input:** Minibatch size $|\mathcal{M}_0|$ and $|\mathcal{M}|$, and total number of rounds $T$
 1: Initialize $\mathbf{x}^0 = 0$;
 2: **for** $t = 1$ **to** $T$ **do**
 3:    **if** $t = 1$ **then**
 4:       Sample a minibatch $\mathcal{M}_0$ of $(\gamma, \mathbf{z})$ according to $p(\mathbf{z}, \gamma; \mathbf{x}^0)$ and compute $\mathbf{g}^0$ using (41);
 5:    **else**
 6:       Compute the Hessian approximation $\tilde{\nabla}^2_{\mathcal{M}} = \frac{1}{|\mathcal{M}|} \sum_{k=1}^{|\mathcal{M}|} \tilde{\nabla}^2_k$ corresponding to $\mathcal{M}$ according to (23);
 7:       Construct $\tilde{\Delta}^t$ based on (13);
 8:       Update the stochastic gradient approximation $\mathbf{g}^t := \mathbf{g}^{t-1} + \tilde{\Delta}^t$;
 9:    **end if**
10:    Compute the ascent direction $\mathbf{v}^t := \operatorname{argmax}_{\mathbf{v} \in \mathcal{C}} \{\mathbf{v}^\top \mathbf{g}^t\}$;
11:    Update the variable $\mathbf{x}^{t+1} := \mathbf{x}^t + 1/T \cdot \mathbf{v}^t$;
12: **end for**

---

## 8.5 Detailed Implementation of SCG++ for Multilinear Extension

While we briefly mentioned the Hessian estimator $\tilde{\nabla}^2_k$ in (23). In this section, we describe SCG++ for the Multilinear Extension problem (21) in Algorithm 2. In particular, we specify the gradient construction for $\mathbf{x}^0$ use the fact that

$$[\nabla F(\mathbf{x})]_i = F(\mathbf{x}; \mathbf{x}_i \leftarrow 1) - F(\mathbf{x}; \mathbf{x}_i \leftarrow 0), \tag{40}$$

for the multilinear extension $F$ [22]. Since both terms in (40) are expectation, we can directly sample a mini-batch $\mathcal{M}_0$ of $(\gamma, \mathbf{z})$ pair from (40) to obtain an unbiased estimator of $\nabla F(\mathbf{x})$ by

$$[\mathbf{g}^0]_i \overset{\text{def}}{=} \frac{1}{|\mathcal{M}_0|} \sum_{k=1}^{|\mathcal{M}_0|} f_{\gamma_k}(N(\mathbf{z}_k) \cup \{i\}) - f_{\gamma_k}(N(\mathbf{z}_k) \setminus \{i\}). \tag{41}$$

## 8.6 Multilinear extension Hessian

In the following lemma, we first study the structure of the Hessian of the objective function for the problem in (22).

**Lemma 4.** *[6] Recall the definition of $F$ in* (22) *as the multilinear extension of the set function $f$ defined in* (20). *Then, for $i = j$ we have $[\nabla^2 F(\mathbf{y})]_{i,j} = 0$, and for $i \neq j$ we have*

$$[\nabla^2 F(\mathbf{y})]_{i,j} = F(\mathbf{y}; \mathbf{y}_i \leftarrow 1, \mathbf{y}_j \leftarrow 1) - F(\mathbf{y}; \mathbf{y}_i \leftarrow 1, \mathbf{y}_j \leftarrow 0)$$
$$- F(\mathbf{y}; \mathbf{y}_i \leftarrow 0, \mathbf{y}_j \leftarrow 1) + F(\mathbf{y}; \mathbf{y}_i \leftarrow 0, \mathbf{y}_j \leftarrow 0), \tag{42}$$

*where the vector $\mathbf{y}; \mathbf{y}_i \leftarrow c_i, \mathbf{y}_j \leftarrow c_j$ is defined as a vector that the $i^{th}$ and $j^{th}$ entries of $\mathbf{y}$ to $c_i$ and $c_j$, respectively.*

*Proof.* First note that

$$\nabla_{\mathbf{x}_i} \log Ber(\mathbf{z}_i; \mathbf{x}_i) = \frac{\mathbf{z}_i}{\mathbf{x}_i} - \frac{1 - \mathbf{z}_i}{1 - \mathbf{x}_i}. \tag{43}$$

We use $\mathbf{z}_{\backslash i,j}$ to denote the random vector $\mathbf{z}$ excluding the $i^{th}$ and $j^{th}$ entries, and denote $\mathbf{z}; \mathbf{z}_i \leftarrow c_i, \mathbf{z}_j \leftarrow c_j$ as the random vector obtained by setting the $i^{th}$ and $j^{th}$ entries of $\mathbf{z}$ to corresponding constants $c_i$ and $c_j$. Compute $\mathbb{E}_{\mathbf{z} \sim p(\mathbf{z}; \mathbf{x})} [\tilde{\nabla}^2 F(\mathbf{y}; \mathbf{z}, \gamma)]_{i,j}$ using (39)

$$\mathbb{E}_{\mathbf{z} \sim p(\mathbf{z}, \gamma; \mathbf{x})} [\tilde{\nabla}^2 F(\mathbf{y}; \mathbf{z}, \gamma)]_{i,j} = \mathbb{E}_{\mathbf{z} \sim p(\mathbf{z}; \mathbf{x})} \left[ f(N(\mathbf{z})) [\nabla_{\mathbf{x}_i} \log Ber(\mathbf{z}_i; \mathbf{x}_i)] [\nabla_{\mathbf{x}_j} \log Ber(\mathbf{z}_j; \mathbf{x}_j)] \right]$$
$$= \sum_{c_i, c_j \in \{0,1\}^2} \mathbb{E}_{\mathbf{z}_{\backslash i,j}} f(N(\mathbf{z}; \mathbf{z}_i \leftarrow c_i, \mathbf{z}_j \leftarrow c_j)) (-1)^{c_i} (-1)^{c_j}$$

where in the first equality we use $\mathbb{E}_\gamma f_\gamma = f$ and the second one uses

$$\mathbf{x}_i^{c_i} \cdot (1 - \mathbf{x}_i)^{1-c_i} \cdot \left[ \frac{\mathbf{c}_i}{\mathbf{x}_i} - \frac{1 - \mathbf{c}_i}{1 - \mathbf{x}_i} \right] = -(-1)^{c_i}. \tag{44}$$

We discuss in detail the configuration of $c_i = c_j = 1$. The other three configurations can be obtained similarly.

$$\mathbb{E}_{\mathbf{z}_{\setminus i,j}} f(N(\mathbf{z}; \mathbf{z}_i \leftarrow 1, \mathbf{z}_j \leftarrow 1)) = F(\mathbf{y}; \mathbf{y}_i \leftarrow 1, \mathbf{y}_j \leftarrow 1), \tag{45}$$

which recovers the first term in (42). □

## 8.7 Proof of Theorem 2

Under bounded marginal value assumption in Theorem 2, the $\| \cdot \|_{2,\infty}$ norm of the Hessian estimator $\tilde{\nabla}_k^2$ has bounded second-order moment:

$$\mathbb{E}\|\tilde{\nabla}_k^2\|_{2,\infty}^2 = \mathbb{E}(\max_{i \in [d]} \|\tilde{\nabla}_k^2(:, i)\|^2) \leq 4d \cdot \mathbb{E}_\gamma D_\gamma^2 = 4d \cdot D_f^2.$$

To prove the claim in Theorem 2 we first prove the following lemma. The following lemma exploits the sparsity of $\mathbf{v}^t$ and the above bound to give a tighter variance bound on $\mathbf{g}^t$ with an explicit dependence on the problem dimension $d$.

**Lemma 5.** *Recall the constructions of the gradient estimator* (41) *and the Hessian estimator* (23). *In the multilinear extension problem* (21)*, under the bounded marginal value assumption in Theorem 2, we have the following variance bound*

$$\mathbb{E}\|\mathbf{g}^t - \nabla F(\mathbf{x}^t)\|^2 \leq \frac{4r^2 d \cdot \epsilon}{|\mathcal{M}|} D_f^2 + \frac{dD_f^2}{|\mathcal{M}_0|}. \tag{46}$$

*Proof.* For $k = 0$, we bound

$$\mathbb{E}_{\mathcal{M}_0}\|\mathbf{g}^0 - \nabla F(\mathbf{x}^0)\|^2 \leq \frac{1}{|\mathcal{M}_0|} \sum_{i=1}^d D_f^2 = \frac{dD_f^2}{|\mathcal{M}_0|}. \tag{47}$$

Let $\tilde{\nabla}_1^2$ be one of the i.i.d. summands in $\tilde{\nabla}_t^2$.

$$\begin{aligned}
&\mathbb{E}\|\mathbf{g}^t - \nabla F(\mathbf{x}^t)\|^2 \\
&= \mathbb{E}\|\tilde{\Delta}^t + \mathbf{g}^{t-1} - \nabla F(\mathbf{x}^t)\|^2 \\
&= \mathbb{E}\|\tilde{\Delta}^t - (\nabla F(\mathbf{x}^t) - \nabla F(\mathbf{x}^{t-1}))\|^2 + \mathbb{E}\|\mathbf{g}^{t-1} - \nabla F(\mathbf{x}^{t-1})\|^2 \\
&= \frac{1}{|\mathcal{M}|}\mathbb{E}\|\tilde{\nabla}_1^2[\mathbf{x}^t - \mathbf{x}^{t-1}] - (\nabla F(\mathbf{x}^t) - \nabla F(\mathbf{x}^{t-1}))\|^2 + \mathbb{E}\|\mathbf{g}^{t-1} - \nabla F(\mathbf{x}^{t-1})\|^2 \\
&\leq \frac{1}{|\mathcal{M}|}\mathbb{E}\|\tilde{\nabla}_1^2[\mathbf{x}^t - \mathbf{x}^{t-1}]\|^2 + \mathbb{E}\|\mathbf{g}^{t-1} - \nabla F(\mathbf{x}^{t-1})\|^2
\end{aligned}$$

Observe that $\mathbf{x}^{t+1} - \mathbf{x}^t = \epsilon\mathbf{v}^t$ which has $r \epsilon$ entries and $d - r$ 0 entries and therefore

$$\begin{aligned}
\mathbb{E}\|\mathbf{g}^t - \nabla F(\mathbf{x}^t)\|^2 &\leq \frac{r^2\epsilon^2}{|\mathcal{M}|}\mathbb{E}\|\tilde{\nabla}_1^2\|_{2,\infty}^2 + \mathbb{E}\|\mathbf{g}^{t-1} - \nabla F(\mathbf{x}^{t-1})\|^2 \\
&\leq \frac{4r^2 d\epsilon^2}{|\mathcal{M}|}D_f^2 + \mathbb{E}\|\mathbf{g}^{t-1} - \nabla F(\mathbf{x}^{t-1})\|^2.
\end{aligned}$$

Repeat the above recursion $t \leq \frac{1}{\epsilon}$ times, we obtain

$$\mathbb{E}\|\mathbf{g}^t - \nabla F(\mathbf{x}^t)\|^2 \leq \frac{4r^2 d \cdot t \cdot \epsilon^2}{|\mathcal{M}|}D_f^2 + \frac{dD_f^2}{|\mathcal{M}_0|} \leq \frac{4r^2 d \cdot \epsilon}{|\mathcal{M}|}D_f^2 + \frac{dD_f^2}{|\mathcal{M}_0|} \tag{48}$$

□

*Proof.* (Proof of Theorem 2) From calculus, we know that

$$|F(\mathbf{x}^{t+1}) - F(\mathbf{x}^t) - \langle \nabla F(\mathbf{x}^t), \mathbf{x}^{t+1} - \mathbf{x}^t \rangle|$$

$$= \frac{1}{2}\int_0^1 |\langle \nabla^2 F(\mathbf{x}(a))(\mathbf{x}^{t+1} - \mathbf{x}^t), (\mathbf{x}^{t+1} - \mathbf{x}^t) \rangle| \mathbf{d}a$$

$$\leq \frac{1}{2}\int_0^1 \|\nabla^2 F(\mathbf{x}(a))(\mathbf{x}^{t+1} - \mathbf{x}^t)\| \cdot \|\mathbf{x}^{t+1} - \mathbf{x}^t\| \mathbf{d}a \qquad (49)$$

$$\overset{(i)}{\leq} \frac{1}{2}\int_0^1 \sqrt{r \cdot \|\nabla^2 F(\mathbf{x}(a))\|_{2,\infty}^2} \cdot \|\mathbf{x}^{t+1} - \mathbf{x}^t\|^2 \mathbf{d}a$$

$$\leq \frac{1}{2}\sqrt{rdD_f^2} \cdot \|\mathbf{x}^{t+1} - \mathbf{x}^t\|^2,$$

where $\mathbf{x}(a) = a\mathbf{x}^t + (1-a)\mathbf{x}^{t-1}$ with $0 \leq a \leq 1$ and we use $\mathbf{x}^{t+1} - \mathbf{x}^t = 1/T \cdot \mathbf{v}^t$ which has $r$ non-zero entries and $d - r$ 0 entries in inequality (i). We thus have the following bound on $F(\mathbf{x}^{k+1})$:

$$F(\mathbf{x}^{t+1}) \geq F(\mathbf{x}^t) + \langle \nabla F(\mathbf{x}^t), \mathbf{x}^{t+1} - \mathbf{x}^t \rangle - \sqrt{rdD_f^2}\|\mathbf{x}^{t+1} - \mathbf{x}^t\|^2$$

$$= F(\mathbf{x}^t) + \frac{1}{T}\langle \nabla F(\mathbf{x}^t) - \mathbf{g}^t, \mathbf{v}^t \rangle + \frac{1}{T}\langle \mathbf{g}^t, \mathbf{v}^t \rangle - \frac{\sqrt{rdD_f^2}}{T^2}\|\mathbf{v}^t\|^2$$

$$\geq F(\mathbf{x}^t) + \frac{1}{T}\langle \nabla F(\mathbf{x}^t) - \mathbf{g}^t, \mathbf{v}^t \rangle + \frac{1}{T}\langle \mathbf{g}^t, \mathbf{x}^* \rangle - \frac{\sqrt{rdD_f^2}}{T^2}\|\mathbf{v}^t\|^2$$

Take expectation on both sides and use the unbiasedness of $\mathbf{g}^t$ to yield

$$\mathbb{E}F(\mathbf{x}^{t+1}) \geq \mathbb{E}F(\mathbf{x}^t) + \frac{1}{T}\mathbb{E}\langle \nabla F(\mathbf{x}^t), \mathbf{x}^* \rangle + \frac{1}{T}\mathbb{E}\langle \nabla F(\mathbf{x}^t) - \mathbf{g}^t, \mathbf{v}^t \rangle - \frac{\sqrt{r^3dD_f^2}}{T^2}. \qquad (50)$$

From the monotonicity of $F$ and the concavity of $F$ along positive directions, we have $\langle \nabla F(\mathbf{x}^t), \mathbf{x}^* \rangle \geq F(\mathbf{x}^*) - F(\mathbf{x}^t)$. Additionally, using the Young's inequality, we write

$$\mathbb{E}F(\mathbf{x}^{t+1}) \geq \mathbb{E}F(\mathbf{x}^t) + \frac{1}{T}\mathbb{E}[F(\mathbf{x}^*) - F(\mathbf{x}^t)] - \frac{1}{2\sqrt{rdD_f^2}}\mathbb{E}\|\nabla F(\mathbf{x}^t) - \mathbf{g}^t\|^2 - \frac{2\sqrt{r^3dD_f^2}}{T^2}.$$

Recall the variance bound (48)

$$\mathbb{E}\|\mathbf{g}^t - \nabla F(\mathbf{x}^t)\|^2 \leq \frac{4r^2d \cdot \epsilon}{|\mathcal{M}|}D_f^2 + \frac{dD_f^2}{|\mathcal{M}_0|}.$$

By choosing $|\mathcal{M}| = \frac{2}{\epsilon}$ and $|\mathcal{M}_0| = \frac{1}{2r^2\epsilon^2}$, we have

$$\mathbb{E}\|\nabla F(\mathbf{x}^t) - \mathbf{g}^t\|^2 \leq 4r^2d \cdot \epsilon^2 D_f^2 \qquad (51)$$

and consequently by setting $T = \frac{1}{\epsilon}$ we have

$$\mathbb{E}F(\mathbf{x}^{t+1}) \geq \mathbb{E}F(\mathbf{x}^t) + \epsilon\mathbb{E}[F(\mathbf{x}^*) - F(\mathbf{x}^t)] - 6\sqrt{r^3dD_f^2} \cdot \epsilon^2,$$

which can be translated to

$$\mathbb{E}[F(\mathbf{x}^*) - F(\mathbf{x}^{\frac{1}{\epsilon}})] \leq (1-\epsilon)^{\frac{1}{\epsilon}}\mathbb{E}[F(\mathbf{x}^*) - F(\mathbf{x}^0)] - 6\sqrt{r^3d} \cdot D_f \cdot \epsilon.$$

In conclusion, we have

$$\mathbb{E}F(\mathbf{x}^{\frac{1}{\epsilon}}) \geq (1 - 1/e)\mathbb{E}[F(\mathbf{x}^*)] - 6\sqrt{r^3d} \cdot D_f \cdot \epsilon.$$

$\square$

## 8.8 Proof of Theorem 3

Consider the classical problem of maximizing a monotone submodular function subject to a cardinality constraint, $\max\{f(S) : |S| \le k\}$. It is known that there exists a monotone submodular set function, denoted by $f_0$, for which obtaining a $(1 - 1/e + \epsilon)$OPT solution requires exponentially many, namely $\exp(\alpha(\epsilon)k)$ for some constant $\alpha(\epsilon) > 0$, function value queries no matter what algorithm is used [24]. To fix the notation, we assume that $f_0$ is defined over the ground set $[n] \triangleq \{1, \cdots, n\}$ and let

$$\text{OPT}(f_0, k) = \max_{S:|S| \le k} f_0(S). \tag{52}$$

For $\delta \in [0, 1/2]$ consider the submodular set function $g_\delta : 2^{[n+2]} \to \mathbb{R}$ defined as follows:

$$g_\delta(S) = \begin{cases} \delta & \text{if } n+1 \in S, \\ \frac{\delta}{2} & \text{if } n+1 \notin S \ \& \ n+2 \in S, \\ 0 & \text{o.w.} \end{cases}$$

Note that the function $g_\delta$ can be defined as an expectation in the following way:

$$g_\delta(S) = \mathbb{E}\left[A\mathbb{1}\{n+1 \in S\} + B\mathbb{1}\{n+1 \notin S \ \& \ n+2 \in S\}\right], \tag{53}$$

where $A, B$ are independent binary random variables given by $A = \text{Bernoulli}(\delta), B = \text{Bernoulli}(\delta/2)$.

Furthermore, we define the submodular function $f_\delta : 2^{[n+2]} \to \mathbb{R}$ as

$$f_\delta(S) = \min\{\text{OPT}(f_0, k), f_0(S \cap [n])\} + \text{OPT}(f_0, k)g_\delta(S). \tag{54}$$

We consider the following maximization problem with the $(k+1)$-cardinality constraint:

$$\text{OPT}(f_\delta, k+1) = \max_{S:|S| \le k+1} f_\delta(S). \tag{55}$$

Note that

$$\text{OPT}(f_\delta, k+1) = (1+\delta)\text{OPT}(f, k). \tag{56}$$

Finally, we consider the following stochastic oracle that, when queried for the function value $f_\delta(S)$, returns the following unbiased estimate: The oracle first computes an unbiased estimate of $g_\delta(S$ by drawing independent samples of variables $A$ and $B$ given in (53), and plugs-in the resulting value into (54) to obtain an unbiased sample for $f_\delta(S)$.

Now, consider an algorithm which aims to maximize $f_\delta$ subject to the $(k+1)$-cardinality constraint (i.e. Problem (55)) by assuming only access to the stochastic oracle mentioned above. Note here that the algorithm does not have any prior information about the structure of the function $f_\delta$, and the only information that it obtains is through the stochastic oracle. In particular, the algorithm does not know a priori $g_\delta(\{n+1\})$ is larger than $g_\delta(\{n+2\})$.

In order to obtain a $(1 - 1/e - \frac{\delta}{4})$-optimal solution for this problem, the algorithm has to either find a $(1 - 1/e + \frac{\delta}{4})$-optimal solution to Problem (52), or it has to know that $g_\delta(\{n+1\})$ is larger than $g_\delta(\{n+2\})$. The former case needs at least $\exp \alpha(\delta/8)k$ queries from the oracle, and the latter case needs at least $O(1/\delta^2)$ oracle queries since it is equivalent to the problem of distinguishing between two Bernoulli random variables $A = \text{Bernoulli}(\delta)$, and $B = \text{Bernoulli}(\delta/2)$–see Lemma 3 in [1].