[Reviews · NeurIPS 2019]

Reviewer 1



The results established in this work are significant from a theoretical standpoint. Through the proposed algorithm SCG++ , the authors have developed a method where quite literally both the upper and lower bounds match for maximizing a monotone DR-submodular function 'F', being true to title of the paper. The approach taken in proving this result based on a variance reduction method that estimates the difference of gradients in the non-oblivious stochastic setting is novel. The ideas developed in Section 4 are very interesting and useful. So on originality, quality and significance, I would rate the paper in the top tier. My primary concern with the paper is how much these results add value in a practical setting. Since no experimental evidence are provided, it is hard to judge the computational speed up gained by SCG++ over competing methods like SCG, SGA, Continuous Greedy methods etc. when applied to an actual problem. For instance: a) When both SGA, SCG and SCG++ use the same number of gradients, what is the difference between the function values at the obtained solution? b) Experimentally, how many number of gradient evaluations does each of these algorithms need to obtain the same quality of the solution? c) In the setting when the monotone DR-submodular function 'F' is available in closed form and the its gradients can be computed exactly, how much the variance reduction method based on difference of gradients actually help say in reducing the variance and giving a superior estimate? Disclaimer: I have not read the proof in detail to aver its accuracy. Typo: 1. In line 120, I believe it should be \nabla f(y, \mathcal{M}). \nabla is missing before 'f'. EDIT: ----- The authors have addressed my concerns in their rebuttal report.

Reviewer 2



High level takeaway from this paper: This is a solid theoretical work with improved theoretical and algorithmic results. Pros of this paper: - Tighter approximation guarantee while achieving a tight convergence rate - Improved variance reduction technique for estimating differences of gradients which is a critical ingredient in their setting - Lower bounds showing their results are tight - Extension to stochastic submodular maximization Cons of this work (also includes some questions/clarifications which are not clear in the paper) - Seems like a condensed version of a much longer paper: While this is not uncommon in NeurIPS, it does affect the clarity of the paper. Also all the proofs and main ideas are in the extended version which makes the paper harder to go through. This is not a criticism as such, but just a suggestion to improve the readability of this manuscript - Extension to the discrete setting: While the result seems to follow from the results of the continuous submodular counterpart and the multilinear extension, I do not understand how one would compute the multilinear extension efficiently. One still needs to sample the ML extension which is high polynomial in complexity? I'm not sure how the authors are circumventing this. - Lack of concrete examples: This comment is coming more from a practitioners perspective. It is unclear how to use such an algorithm in practice. What are concrete examples of continuous submodular functions and how do the results in this paper impact them? Lack of empirical results demonstrating the improved convergence: Related to the above. This paper has no empirical results to demonstrate how these carry over in practice. Does the improved convergence results imply improved performance in real world applications? I would like to see this in the main version of the paper.

Reviewer 3



Originality: The paper studies a known problem of maximizing a continuous submodular function under convex constraints. The main technique to achieve the improved number of queries are 1) adapt the gradient variance reduction technique to the non-oblivious case using Hessian and 2) the approximation of the Hessian with O(d) gradient evaluations. The proposed bound needs the assumption about the smoothness of the Hessian (assumption 4.7), and I think without such an assumption, it would require O(d^2 \epsilont^-2) oracle calls. The dependence on d seems also important to me. Maybe the authors can illustrate more on this point. Quality: The paper has theoretical analysis of the proposed algorithm as well as a lower bound. Clarity: The paper is well organized. Significance: The proposed algorithm achieves the best-known results of the number of queries to get an almost optimal guarantee for DR submodular functions. The number of queries is also shown to match the information theoretical lower bound. The paper's result is therefore significant. After rebuttal: I have read all other reviewers' comments and the authors' feedback. I keep my evaluations unchanged.

[Author Response · NeurIPS 2019]

We thank the reviewers for their careful consideration and constructive feedback. Below, please find our responses.

**General response to all reviewers regarding empirical study of SCG++.**

This paper is primarily of theoretical nature and aims to provide a complete an-
swer to an open question regarding the sample complexity of DR-Submodular
maximization with stochastic oracles. However, the main suggestion of all the
reviewers for improving the paper has been to provide an empirical study on
SCG++ and compare its performance/complexity with the existing methods
such as stochastic continuous greedy (SCG). We will hence provide a separate
section in the revised manuscript that provides a thorough empirical study on
the performance of SCG++ by comparing it with the state-of-the-art methods
using practical applications. Moreover, in order to highlight the superiority of
our method (SCG++) compared to the existing methods, we have implemented
both SCG++ and SCG on a real-world instance of (stochastic) discrete submodular optimization (see equation (20) in
the paper). The setting involves selecting a subset of movies to recommend to users based on previously obtained data
(ratings) from the users (we have used the MovieLens data set with 1M entries). In the plot provided above, we let the
x-axis (resp. y-axis) represents the amount of function evaluations that SCG++ (resp. SCG) needs to achieve the *same*
solution quality. In other words, every point in the plot corresponds to a specific solution quality and represents the total
number of function evaluations needed by SCG++ (the x-coordinate of the point) as well as SCG (the y-coordinate) to
achieve that solution quality. As we observe from the plot, SCG requires a higher number of function evaluations with
respect to SCG++ and the difference diverges as the solution quality increases.

**Response to Reviewer #1. Q1:** My primary concern with the paper is how much these results add value in a practical
setting. Since no experimental evidence are provided... **A1:** Please see our general response above. We would also
like to emphasize that, as proven in the paper, SCG++ beats the existing methods in every aspect. Hence, we expect to
observe a similar superiority in our numerical simulations (we have provided one example in the plot above).

**Q2:** If 'F' is available in closed form and the its gradients can be computed exactly.... **A2:** If the gradients can be
computed exactly then no variance reduction is needed (i.e. if there is no stochaticity/randomness, then there is no
variance to be reduced). In this case, both SCG and SCG++ reach a $(1 - 1/e - \epsilon)$-opt solution with $O(1/\epsilon)$ oracle calls.

**Response to Reviewer #2. Q3:** Improve the readability. **A3:** We will provide further explanation in the main part of
the revised version about the ideas behind our methods and will include proof sketches. Thanks for your suggestions.

**Q4:** Extension to the discrete setting: I do not understand how one would compute the multilinear extension efficiently.
**A4:** We do not need to compute the exact value of the multilinear extension; We only need an unbiased estimate of the
value of the multilinear extension which can easily be obtained using a single function evaluation. More precisely, the
multilear extension is defined as $F(\mathbf{x}) = \mathbb{E}_{S \sim \mathbf{x}}[f(S)]$ and an unbiased estimate of $F(\mathbf{x})$ can be obtained by generating
a random subset $S \sim \mathbf{x}$ and computing the corresponding function value $f(S)$ as the estimate. In a similar manner,
we can obtain an unbiased estimate of the partial derivatives as well as the entries of the Hessian of the multilinear
extension using a few function evaluations (as explained in detail in lines L272-280 of the paper). These unbiased
estimates are the plugged into our proposed stochastic algorithms to provide the desired solutions. Theorem 2 provides
the exact sample complexity (i.e. the number of function evaluations) of our proposed algorithms. Also, Sections
8.4-8.6 in the appendix provide a detailed explanation on how to implement our methods in the discrete setting).

**Q5:** Lack of concrete examples: What are concrete examples of continuous submodular functions and how do the
results in this paper impact them? **A5:** We will address this comment thoroughly in our revised version. In brief,
continuous submodular functions generalize the notion of diminishing returns to continuous domains. They have
recently gained considerable attention and have been used in applications such as inference in determinental point
processes, resource allocation, experimental design, etc. Moreover, the multilinear extension of any discrete submodular
function is a continuous submodular function. The main impact of our algorithms is in the so-called stochastic setting:
Oftentimes in practice, the function value (or its gradient) can not be computed exactly and we need to consider
(stochastic) estimates. For example, in large scale-settings, computing the function (or its gradient) could amount to a
full pass over the data which is an expensive task and hence we resort to stochastic min-batches to speed up computation
instead of a full pass over data. Indeed, stochastic optimization techniques are among the key contributors to the success
of large-scale machine learning. Our paper provides algorithms for stochastic submodular optimization with optimal
sample complexity (i.e. the minimum number of queries from the stochastic oracle).

**Q6:** Lack of empirical results demonstrating the improved convergence. **A6:** Please check our general response above.

**Response to Reviewer #3.** We would like to thank the reviewer for the comments and suggestions. About the empirical
experiments, please see the general response given above.

**Q7:** The proposed bound needs the smoothness of the Hessian , and I think without such an assumption, it would
require $O(d^2/\epsilon^2)$ oracle calls. **A7:** The reviewer is right that if we do not assume Hessian smoothness, then for running
SCG++ we need overall $1/\epsilon^2$ stochastic Hessian approximations which means that the overall computational complexity
of SCG++ will be $d^2/\epsilon^2$. We will highlight this point in the revised version.

[Meta-Review · NeurIPS 2019]

This paper considers DR-submodular maximization under convex constraints. It achieves optimal results in terms of both approximation and query complexity. The first paper in this line of work shows that Stochastic Gradient Ascent (SGA) achieves a 1/2-epsilon approximation in 1/epsilon^2 stochastic gradient. There have been multiple papers on this problem since then. This paper introduces SCG++ which achieves a 1-1/e-epsilon in 1/epsilon^2 stochastic oracle calls using a novel variance reduction method. The authors show a lower bound, proving that this bound is optimal. In addition, the authors also point out corollaries for maximizing a noisy sub modular functions set function subject to matroid constraints.